# CTRLA: ADAPTIVE RETRIEVAL-AUGMENTED GENERATION VIA INHERENT CONTROL

## ABSTRACT

Retrieval-augmented generation (RAG) has emerged as a promising solution for mitigating hallucinations of large language models (LLMs) with retrieved external knowledge. Adaptive RAG enhances this approach by enabling dynamic retrieval during generation, activating retrieval only when the query exceeds LLM's internal knowledge. Existing methods primarily focus on detecting LLM's confidence via statistical uncertainty. Instead, we present the first attempts to solve adaptive RAG from a representation perspective and develop an inherent control-based framework, termed CTRLA. Specifically, we extract the features that represent the honesty and confidence directions of LLM and adopt them to control LLM behavior and guide retrieval timing decisions. We also design a simple yet effective query formulation strategy to support adaptive retrieval. Experiments show that CTRLA is superior to existing adaptive RAG methods on a diverse set of tasks. Honesty steering can effectively make LLMs more honest and confidence monitoring is a promising indicator of retrieval trigger. Our anonymous codes are submitted with the paper and will be publicly available.

## 1 INTRODUCTION

Retrieval-augmented generation (RAG; Guu et al. 2020; Izacard et al. 2023) has proven effective in mitigating hallucination by integrating external knowledge into LLMs. Early efforts often employ single-round, indiscriminate retrieval, resulting in over-reliance on external knowledge and incomplete retrieval (Wang et al., 2023; Su et al., 2024a). To solve the issues, adaptive RAG (ARAG; Jiang et al. 2023b; Wang et al. 2024a) has emerged, which enables dynamic retrieval during generation, activating retrieval only when the query exceeds LLM's internal knowledge (Ni et al., 2024).

The key challenges in ARAG involve determining *what* and *when* to retrieve (Su et al., 2024b; Yao et al., 2024). The design of *what* aspect typically depends on the construction of *when* aspect, making ARAG's primary focus the issues related to *when* aspect. For the *when* aspect, recent ARAGs leverage the ability that LLMs are aware of their uncertainty (Kuhn et al., 2023; Chen et al., 2024; Xiong et al., 2024), utilizing this characteristic to determine retrieval timing by assessing *confidence* level of their knowledge (Su et al., 2024b; Yao et al., 2024). They primarily focus on detecting uncertainty in the LLM's outputs to signal retrieval, relying on factors such as output probabilities (Jiang et al., 2023b), entropy of output (Su et al., 2024b) or internal states (Yao et al., 2024), or verbal feedback (Wang et al., 2024b; Yan et al., 2024). From a statistical standpoint, uncertainty and confidence are conceptually equivalent, both reflect the degree of certainty in a model's predictions (Yang et al., 2023; Band et al., 2024; Tao et al., 2024). Thus, uncertainty can act as a proxy for confidence when determining retrieval timing.

We revisit the assumptions underlying these uncertainty-based methods. First, they presume that LLM's output aligns with its internal knowledge (Lin et al., 2022; Zou et al., 2023), that is, LLM can accurately reflect its internal knowledge in outputs, *i.e.,* they are *honest*. However, LLMs often navigate a trade-off between honesty and helpfulness, balancing between discerning its limitations and generating user-satisfied plausible content (Liu et al., 2024a). When the output diverges from internal knowledge, indicating low honesty, they only detect intended output rather than internal knowledge. Second, they equate uncertainty with LLM's *confidence*, [1] which may be not always

---

[1] Confidence is the feeling of belief or trust that a person or thing is reliable (Bandura, 1997).

applicable to LLM behavior. For instance, an LLM may frequently respond with "I don't know" or "insufficient information," suggesting low uncertainty, yet retrieval should still occur. Moreover, semantically equivalent answers can be expressed in various ways in free-form generation, which may lead to high uncertainty (Farquhar et al., 2024). However, retrieval is unnecessary in this scenario.

Based on this analysis, we emphasize both *honesty* and *confidence* of LLMs are crucial for accurate retrieval timing. However, current ARAGs struggle to address them due to the limitations of statistical uncertainty. We propose to solve ARAG from a representation perspective (Olah, 2023; Bricken et al., 2023; Zou et al., 2023; Templeton et al., 2024), developing an efficient and unified framework that seamlessly tackles the requirements of honesty and confidence. Our core idea involves extracting features corresponding to *honesty* and *confidence* directions from LLMs and using them to control LLM behavior and guide retrieval timing decisions simultaneously.

We devise an Inherent **Co**nt**r**ol-based **A**daptive RAG framework (**CTRLA**). To steer LLM toward honesty and monitor its confidence, we extract features aligned with the directions of honesty and confidence within LLM's representation space. By adjusting the honesty direction—a process we refer to as *honesty steering*—we can shift the LLM's representation space to promote more honesty outputs. Simultaneously, confidence is quantified by measuring the projection of current representation onto the confidence feature, a method we call *confidence monitoring*. Honesty steering helps LLM recognize its limitations and suppress the generation of fabricated plausible information. Confidence monitoring, in turn, enhances the precision of retrieval timing. We also implement a simple yet effective query formulation module to support adaptive retrieval, minimizing the impact of noise and intent drift. Extensive experiments verify the effectiveness of CTRLA, revealing that adjusting the directions of LLM's internal states enhances its honesty, while confidence monitoring reliably signals when to trigger retrieval, optimizing the balance between retrieval and internal knowledge use.

## 2 RELATED WORK

### 2.1 RETRIEVAL-AUGMENTED GENERATION

Early RAG efforts (Lewis et al., 2020; Karpukhin et al., 2020; Zhu et al., 2021; Komeili et al., 2022; Khattab et al., 2023) relied on single-round, indiscriminate retrieval, increasing computational costs and degrading model performance (Wang et al., 2023; Su et al., 2024a). To address these issues, ARAG emerged, enabling dynamic retrieval during generation when the query exceeds LLM's internal knowledge (Jiang et al., 2023b; Wang et al., 2024a; Ni et al., 2024). Previous implementations utilized static rules, such as prior sentences (Trivedi et al., 2023), sliding windows (Borgeaud et al., 2022; Ram et al., 2023), and in-context learning (Zhao et al., 2023; Zhang et al., 2024; Li et al., 2024). Recent ARAGs leverage LLMs' self-awareness of uncertainty to optimize retrieval timing by assessing confidence levels through internal states (Yao et al., 2024), likelihoods (Jiang et al., 2023b; Wang et al., 2024a; Su et al., 2024b), or verbal feedback (Wang et al., 2024b; Ding et al., 2024; Yan et al., 2024). This enhances retrieval timing and balances external and internal knowledge use. However, uncertainty-based ARAGs face challenges with LLM honesty and confidence, crucial for accurate retrieval timing. CTRLA addresses these issues from a representation perspective, enhancing control over honesty and confidence to improve retrieval-augmented generation effectiveness.

### 2.2 LINEAR REPRESENTATIONS IN LLMS

Recent research has explored LLM representations to understand their beliefs, interpretability, and compliance (Levinstein & Herrmann, 2023; Li et al., 2023; Bricken et al., 2023). Grounded in the linear representation and superposition hypotheses, these studies suggest that specific features can be aligned with particular directions in the LLMs' linear space. This framework effectively guides and monitors model outputs (Olah, 2023). Researchers have modified or detected models' demeanor, preferences, stated goals, and biases, as well as induced errors or mitigated risks (Templeton et al., 2024). Supporting the hypotheses, Marks & Tegmark (2023) and Slobodkin et al. (2023) found that features like truthfulness and answerability are linearly separable within the latent space. Further efforts (Zou et al., 2023; Liu et al., 2024b) utilized contrastive instruction templates to clarify feature directions. CTRLA leverages these insights to extract features related to honesty and confidence, aiming to control LLM behavior and guide retrieval timing decisions, bridging representational understanding and practical applications.

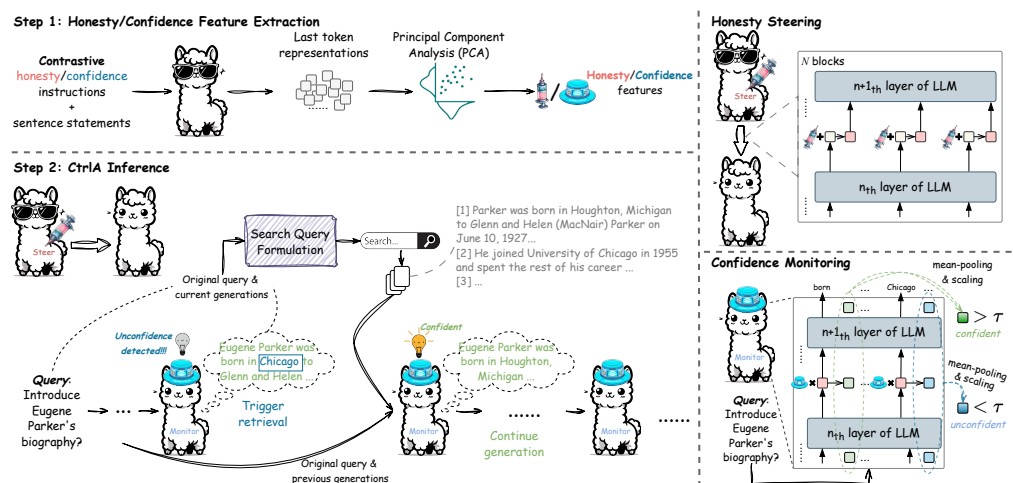

Figure 1: CTRLA framework. Step 1 extracts the features corresponding to *honesty* and *confidence* directions; Step 2 utilizes extracted features to steer and monitor LLM behaviors at inference. The *honesty* feature **steers** the representation of LLM to make it more honest, while *confidence* feature is used to **monitor** the confidence level of LLM outputs, where the token whose score is lower than the threshold is marked as unconfident. The retrieval is triggered if specific tokens are unconfident.

## 3 INHERENT CONTROL BASED ADAPTIVE RAG

### 3.1 PRELIMINARY

Given a query $q$, RAG aims to assist LLMs in generating more precise answers $y = [s_1, \dots, s_m] = [w_1, \dots, w_n]$ containing $m$ sentences or $n$ tokens by retrieving relevant documents $\mathcal{D}_q = \mathcal{R}(q)$ from document corpus $\mathcal{D} = \{d_i\}_{i=1}^{|\mathcal{D}|}$ or web via retriever $\mathcal{R}$. The retrieved documents $\mathcal{D}_q$ are usually concatenated with input $x$, *i.e.,* query $q$ with task instruction $\mathcal{I}$, to aid answer generation as $y = \text{LLM}([\mathcal{D}_q; x])$, where $[\cdot; \cdot]$ denotes concatenation. In contrast, adaptive RAG (Jiang et al., 2023b) performs active retrieval necessity decision via a trigger mechanism $\mathcal{T}(x, y_{<t})$, where $y_{<t}$ is the prior generations as of step $t(t \geq 1)$. If $\mathcal{T}$ is triggered, the query formulation function $q_t = f_q(x, y_{<t})$ will produce a query $q_t$ to search. If $\mathcal{T}$ is triggered at $t = 1$, *i.e.,* $y_{<1} = \emptyset$, $q$ will be original query. Given the retrieved documents $\mathcal{D}_{q_t}$, the model continues generating next output segment (usually, a sentence) $y_t = \text{LLM}([\mathcal{D}_{q_t}; x; y_{<t}])$ till the answer comes to its end or next retrieval trigger occurs.

### 3.2 CTRLA FRAMEWORK

#### 3.2.1 LINEAR REPRESENTATION FEATURE EXTRACTION

Our approach builds on the linear representation and superposition hypotheses (Olah, 2023; Bricken et al., 2023; Templeton et al., 2024). We aim to extract features that represent *honesty* and *confidence* directions from LLM's representation space and use them to steer or monitor its behavior. Specifically, we manually craft contrastive instructions, as shown in Prompt 3.1, to extract features that represent the directions of honesty and confidence. Let $\mathcal{I}_{h/c}^+$ denote the positive instruction of honest or confident, $\mathcal{I}_{h/c}^-$ be the negative instruction of dishonest or unconfident, and $\mathcal{S} = \{s_1, \dots, s_{|\mathcal{S}|}\}$ represent the dataset with $|\mathcal{S}|$ statements used to extract target features.

For honesty feature extraction, each statement $s_i$ is concatenated with both positive and negative instructions, forming $\mathcal{I}_h^+ \oplus s_i$ and $\mathcal{I}_h^- \oplus s_i$, respectively, resulting in $|\mathcal{S}|$ statement pairs. For the statement pair of $s_i$, they are sequentially fed into LLM in a teacher-forcing manner to collect token representations. Given that each LLM layer encodes a unique semantic space (Chuang et al., 2024b; Sun et al., 2024), we extract token representations from all LLM layers. Assuming LLM has $L$ layers and $s_i$ contains $n$ tokens, we obtain representations $\{\{r_{i,k}^{l,+}\}_{k=1}^n\}_{l=1}^L$ and $\{\{r_{i,k}^{l,-}\}_{k=1}^n\}_{l=1}^L$ for positive and negative instructions, where $r_{i,k}^l$ denotes the $k$-th token representation of $s_i$ at layer $l$. The contrastive vector for the $k$-th token at $l$-th layer is computed as $v_{i,k}^l = r_{i,k}^{l,+} - r_{i,k}^{l,-}$. Thus, after

processing all tokens of $s_i$, we derive the set of contrastive vectors $\{\{\boldsymbol{v}_{i,k}^l\}_{k=1}^n\}_{l=1}^L$. Since we employ *teacher-forcing* to encode each token's representation—and "honest" and "dishonest" are the only differing descriptions between $\mathcal{I}_h^+$ and $\mathcal{I}_h^-$—the vector $\boldsymbol{v}_{i,k}^l$ captures the honesty direction for the $k$-th token at layer $l$ (Zou et al., 2023). After processing all statements in $\mathcal{S}$, we apply PCA to the collected contrastive vectors $\{\{\{\boldsymbol{v}_{i,k}^l\}_{k=1}^n\}_{l=1}^L\}_{i=1}^{|\mathcal{S}|}$ at each layer $l$, extracting the first principal component as the general honesty direction. This results in a set of honesty direction vectors $\boldsymbol{v}_h = \{\boldsymbol{v}_h^l\}_{l=1}^L$. Note confidence feature extraction also utilizes the same method to derive $\boldsymbol{v}_c = \{\boldsymbol{v}_c^l\}_{l=1}^L$.

---

**Prompt 3.1: Instruction for Honesty and Confidence Feature Extraction**

**[INST]** `Pretend you're a <honest/dishonest> | <confident/unconfident> person` `making statements about the world.` **[/INST]** `<a statement $s_i$>`

---

We use the True-False dataset (Azaria & Mitchell, 2023) as $\mathcal{S}$ for honesty feature extraction, which tests whether LLMs' internal states reflect truthfulness. For confidence, we synthesize confident and unconfident statements using GPT-4 (ref. Appendix C.1) due to the scarcity of available datasets.

### 3.2.2 Honesty Steering

According to the superposition hypothesis, adjusting LLM by moving each token's representation closer to the direction representing the honesty feature during decoding, is a direct way to enhance its honesty (Olah, 2023; Zou et al., 2023; Templeton et al., 2024). To achieve this, we employ a simple linear combination. After extracting the honesty feature, it can be directly used to steer the behavior of the LLM. Assuming the LLM contains $L$ layers, each layer has its corresponding feature. Let $\boldsymbol{v}_h = \{\boldsymbol{v}_h^l\}_{l=1}^L$ denote the honesty feature and $\boldsymbol{R}_k = \{\boldsymbol{r}_k^l\}_{l=1}^L$ represent the token representations for the $k$-th token at each layer. We then apply a linear combination function for honesty steering:

$$\hat{\boldsymbol{R}}_k = \boldsymbol{R}_k + \lambda \cdot \boldsymbol{v}_h = \{\boldsymbol{r}_k^l + \lambda \cdot \boldsymbol{v}_h^l \mid \forall\, l \in [1, \ldots, L]\}, \tag{1}$$

where the coefficient $\lambda$ controls the strength of honesty steering. Because $\boldsymbol{v}_h$ represents the direction that promotes honesty, the "$+$" operator is used in Eq 1. Conversely, to reduce honesty, the "$-$" operator can be employed. As illustrated in Figure 1, honesty steering is applied layer-by-layer and token-by-token during generation. This method is both simple and effective, with minimal impact on inference costs. For brevity, we denote honesty steering as $\hat{\boldsymbol{y}}_t = \mathcal{P}_h(\boldsymbol{y}_t)$ in the following descriptions.

### 3.2.3 Confidence Monitoring as Retrieval Trigger

According to the linear representation hypothesis, an intuitive way to monitor the LLM's confidence during generation is to evaluate how well token representations align with the confidence feature direction in the representation space (Bricken et al., 2023; Zou et al., 2023; Templeton et al., 2024). Given the extracted confidence feature, we utilize it to monitor LLM's confidence during generation. Let $\boldsymbol{R}_k = \{\boldsymbol{r}_k^l\}_{l=1}^L$ represent the $k$-th token's representation at each layer, and $\boldsymbol{v}_c = \{\boldsymbol{v}_c^l\}_{l=1}^L$ denote the confidence feature. Specifically, we compute the confidence score for $k$-th token using the dot product, followed by mean-pooling across layers and a scaling operation for normalization and outlier removal. This produces the confidence score for the $k$-th token as follows:

$$\begin{aligned}
\tilde{m}_k &= \texttt{meanpool}([m_{k,1}, \ldots, m_{k,L}]) = \texttt{meanpool}\big([\boldsymbol{r}_k^{l,\top} \cdot \boldsymbol{v}_c^l]_{l=1}^L\big), \\
\bar{m}_k &= \texttt{scale}([\tilde{m}_0, \ldots, \tilde{m}_k])[-1] - \tau,
\end{aligned} \tag{2}$$

where $\tau$ is the threshold to adjust the sensitivity of confidence monitoring, $\tilde{m}_{<k}$ represents the mean-pooled score of preceding tokens, and the index $-1$ refers to the score of the last token, *i.e.*, $k$-th token. If $\bar{m}_k > 0$, it suggests that the $k$-th token's representational direction leans towards the confidence, indicating that LLM is confident in generating this token. Conversely, if $\bar{m}_k < 0$, LLM is unconfident in generating the $k$-th token. Here we denote confidence monitoring as $\mathcal{P}_c$.

The goal of confidence monitoring is to serve as a reliable detector for accurately determining appropriate retrieval timing (Wu et al., 2024a; Chuang et al., 2024a). For the $t$-th output segment $\hat{\boldsymbol{y}}_t = [w_{t_s}, \ldots, w_{t_e}]$ of the LLM, with confidence scores $[\bar{m}_{t_s}, \ldots, \bar{m}_{t_e}]$ for each token, the retrieval necessity is measured by the confidence scores of specific tokens within $\hat{\boldsymbol{y}}_t$. We only consider the confidence scores of *new information* in $\hat{\boldsymbol{y}}_t'$, *i.e.,* content that has not appeared in the previous

generation and excludes trivial tokens, like stopwords. The retrieval trigger $\mathcal{T}$ activates if any confidence score in $\hat{\boldsymbol{y}}_t'$ satisfies $\bar{m}_k < 0$, where $t_s \leq k \leq t_e$. If $\hat{\boldsymbol{y}}_t'$ contains such tokens, retrieval is triggered, *i.e.,* $\mathcal{T}(\mathcal{P}_c(\hat{\boldsymbol{y}}_t')) ==$ `True`.

Due to the honesty steering, LLM will generate refusal outputs more frequently, since honesty steering can effectively regulate LLM behavior to make it more honest, leading to more frequent generation of non-responsive or refusal outputs. These refusal responses are well-aligned with the LLM's internal beliefs, *i.e.,* LLM is confident in its knowledge limitations, making them challenging to detect by confidence monitoring. To address this issue, we further develop a refusal handling module, which employs a pattern matching function, as a supplement to confidence monitoring, to identify refusal content. The detailed algorithm is presented in Appendix B.3.1.

### 3.2.4 SEARCH QUERY FORMULATION

Once retrieval is triggered, we need to employ a search query to retrieve relevant documents that aid in LLM generation. The construction of effective search queries plays a pivotal role in enhancing retrieval efficiency (Jiang et al., 2023b). We develop two search query formulation strategies.

**Context-Augmented Querying.** Initially, for a query $\boldsymbol{q}$, we prompt the LLM to sequentially generate responses. Once the retrieval is triggered, context-augmented querying (CAQ) will concatenate the query $\boldsymbol{q}$ with the processed output segment $\hat{\boldsymbol{y}}_t$ for retrieval, since using the original query as a supplement can avoid intent drift and improve the effectiveness of retrieval (Jagerman et al., 2023). Besides, the output segment $\hat{\boldsymbol{y}}_t = [w_{t_s}, \ldots, w_{t_e}]$ may contain noise such as unconfident tokens and incorrect contents, we process the sentence by masking out the tokens, which satisfy (i) not appeared in $\boldsymbol{q}$ and previous generations $\boldsymbol{y}_{<t}$, *i.e.,* new information and (ii) unconfident tokens, as:

$$\texttt{mask}(\hat{\boldsymbol{y}}_t) = \left\{ \bar{w} \bigg| \bar{w} = \begin{cases} \emptyset, & \text{if } w \notin \boldsymbol{q} \cup \boldsymbol{y}_{<t} \text{ and } \bar{m}_w < 0 \\ w, & \text{otherwise} \end{cases}, \forall w \in \hat{\boldsymbol{y}}_t \right\}. \tag{3}$$

Thus, the CAQ generates the refined search query as $f_{\text{CAQ}}(\boldsymbol{x}, \hat{\boldsymbol{y}}_t) = [\boldsymbol{q}; \texttt{mask}(\hat{\boldsymbol{y}}_t)]$.

---

**Algorithm 1** CTRLA Inference

**Require:** language model LM, retriever $\mathcal{R}$, document corpus $\mathcal{D}$, honesty steering $\mathcal{P}_h$, query formulator $f_q$, retrieval trigger $\mathcal{T}$, maximal generation length $L_{\max}$, stop generation token `eos`
1: **input:** prompt $\boldsymbol{x}$ ($\mathcal{I}$ and $\boldsymbol{q}$), previous generation $\boldsymbol{Y}_{<t} = \emptyset$
2: **output:** the final response of input $\boldsymbol{Y}$
3: **while** true **do**
4:     LM along with $\mathcal{P}_h$ predicts next segment $\hat{\boldsymbol{y}}_t$ given $(\boldsymbol{x}, \boldsymbol{Y}_{<t})$
5:     $\mathcal{T}$ simultaneously monitors retrieval signal during LLM generates $\hat{\boldsymbol{y}}_t$
6:     **if** $\mathcal{T} ==$ `True` **then**
7:         $\mathcal{R}$ retrieves $\mathcal{D}_q$ from $\mathcal{D}$ via $\boldsymbol{q}_t = f_q(\boldsymbol{q}, \hat{\boldsymbol{y}}_t)$
8:         LM along with $\mathcal{P}_h$ re-predicts next segment $\hat{\boldsymbol{y}}_t$ given $(\boldsymbol{x}, \boldsymbol{Y}_{<t}, \mathcal{D}_q)$
9:     **end if**
10:    Set $\boldsymbol{Y}_{<t} = [\boldsymbol{Y}_{<t}; \hat{\boldsymbol{y}}_t]$
11:    **if** $\boldsymbol{Y}_{<t}[-1] =$ `eos` or $\texttt{len}(\boldsymbol{Y}_{<t})$ reaches $L_{\max}$ **then**
12:        break
13:    **end if**
14: **end while**
15: Set $\boldsymbol{Y} = \boldsymbol{Y}_{<t}$

---

**Targeted Validation Querying.** CAQ directly masks out the noise of the output segment and concatenates it with the original query to form a search query. Yet, off-the-shelf retrievers may prefer a well-formatted query (Karpukhin et al., 2020). Thus, we also develop a targeted validation querying strategy (TVQ), $f_{\text{TVQ}}$. It instructs LLM to produce a search query using original query and current output segment as references (see Prompt B.1). The goal of TVQ is to generate a query to validate the accuracy of current output segment by searching for supporting documents. For simplicity, we use $f_q$ to represent both $f_{\text{CAQ}}$ and $f_{\text{TVQ}}$.

## 3.3 INFERENCE PROCESS

For an input $x$ and preceding generation $Y_{<t}$, the model generates the output segment along with honesty steer $\mathcal{P}_h$ and derives $\hat{y}_t$. Simultaneously, the confidence monitor $\mathcal{P}_c$ is activated to compute the confidence score of each token during generation. We collect the confidence scores of new information $\hat{y}'_t$ to determine retrieval necessity via retrieval trigger $\mathcal{T}$. If retrieval is not required, the model continues predicting the next output segment. Otherwise, we adopt query formulation, $f_q$, to produce a search query $q_t$ and retrieve documents $\mathcal{D}_q$ via retriever $\mathcal{R}$. The retrieved documents $\mathcal{D}_q$, input $x$, and preceding generation $Y_{<t}$ are concatenated to regenerate the current output segment. Algorithm 1 presents an overview of CTRLA inference step. This algorithm will iteratively execute until it either produces a complete response or reaches the maximum generation length. The detailed algorithm with refusal handling is presented in Appendix B.3.2.

## 4 EXPERIMENT SETUP

**Datasets and Evaluation.** For *short-form* QA, we select PopQA (Mallen et al., 2023) and TriviaQA (Joshi et al., 2017). For *long-form* QA, we use ASQA (Stelmakh et al., 2022) and biography generation (Bio; Min et al. 2023). For *multi-hop* QA, we follow Su et al. (2024b) to choose 2WikiMultihopQA (2WMQA; Ho et al. 2020) and HotpotQA (HQA; Yang et al. 2018). For short-form QA, we report the accuracy. For ASQA, we report str-em, Rouge-L (R-L; Lin 2004), MAUVE (mau; Pillutla et al. 2023), EM and F1. Bio is evaluated by FactScore (FS; Min et al. 2023). For multi-hop QA, we report EM and F1. We also evaluate 500 test samples (v04082024) of FreshQA (Vu et al., 2024) and report both relaxed and strict accuracy scores. More details in Appendix C.2 and C.3.

**Implementation and Retrieval Setup.** We select the Mistral-7B (Jiang et al., 2023a) as the backbone of CTRLA and adopt the greedy decoding strategy for all experiments. The $\lambda$ for honesty steer is set to $0.3$ and $\tau$ for confidence monitoring is set to $0.0$. By default, we use BM25 and BGE (Xiao et al., 2024) as our retriever and use the 2018 English Wikipedia corpus as the document source following Jiang et al. (2023b) and Asai et al. (2024). For PopQA and Bio, we follow Self-RAG (Asai et al., 2024) to retrieve additional information from the web to mitigate coverage limitations in the Wikipedia corpus. For the multi-hop QA task, we only use BM25 as the retriever. For FreshQA, we only retrieve from the web to obtain supporting documents. More details in Appendix C.4 and C.5.

**Baselines.** We compare CTRLA with representative RAG baselines: (1) Single-round RAG (SR-RAG), which retrieves documents before generation; (2) Fix-sentence RAG (FS-RAG; Trivedi et al. 2023), which triggers retrieval every sentence and the previous sentence is used as query; (3) Fix-length RAG (FL-RAG; Ram et al. 2023), which triggers retrieval every $n$ tokens and the previous token window is used as query; (4) Query-decompose RAG (QD-RAG; Press et al. 2023; Khattab et al. 2023), which prompts LLMs to generate follow-up queries and trigger retrieval for each query; (5) Adaptive RAGs: FLARE (Jiang et al., 2023b), Self-RAG (Asai et al., 2024), DRAGIN (Su et al., 2024b), SeaKR (Yao et al., 2024), RQ-RAG (Chan et al., 2024) and QC-RAG (Jeong et al., 2024). **For (1)-(4), we reimplement them under the same setting as CTRLA**. More details about the baselines are in Appendix C.6.

Table 1: Overall results of short-form QA. $\diamond$ is our reproduced results. $\ddagger$ denotes results in the corresponding work.

| Method | TriviaQA | PopQA |
|---|---|---|
| wo-RAG$^{\diamond}_{7B}$ | 53.8 | 25.7 |
| SR-RAG$^{\diamond}_{7B}$ | 62.7 | 51.9 |
| FL-RAG$^{\diamond}_{7B}$ | 60.8 | 28.1 |
| FS-RAG$^{\diamond}_{7B}$ | 54.3 | 26.9 |
| QD-RAG$^{\diamond}_{7B}$ | 52.3 | 29.4 |
| FLARE$^{\diamond}_{7B}$ | 72.4 | 48.3 |
| Self-RAG$^{\ddagger}_{7B}$ | 66.4 | 54.9 |
| Self-RAG$^{\ddagger}_{13B}$ | 69.3 | 55.8 |
| RQ-RAG$^{\ddagger}_{7B}$ | - | 57.1 |
| QC-RAG$^{\ddagger}_{11B}$ | 58.2 | - |
| **CTRLA**$_{7B}$ | **76.4** | **61.8** |

## 5 EXPERIMENT RESULTS AND ANALYSIS

### 5.1 MAIN RESULTS

**Performance comparison.** CTRLA demonstrates consistent superiority over the compared approaches across various tasks and evaluation metrics, as evidenced by the results in short-form QA (Table 1), long-form QA (Table 2), multi-hop QA (Table 3), and the FreshQA dataset (Table 4). In each case, CTRLA surpasses fine-tune based methods (*e.g.,* Self-RAG), uncertainty-based methods

(*e.g.,* FLARE and DRAGIN), and rule-based methods (*e.g.,* FL/FS/QD-RAG). Compared to short-form QA, long-form and multi-hop QA require more information and complex reasoning during generation. CTRLA consistently outperforms all baselines on both tasks. FreshQA contains more diverse question types, including never-changing, slow-changing, fast-changing, and false-premise questions, as well as single-hop and multi-hop questions, CTRLA shows strong generalization capability on different question types, leading to better performance than the compared baselines. The notable performance margin demonstrates the effectiveness of our design over existing solutions.

Table 2: Overall results of long-form QA. $\diamond$ is our reproduced results. $\ddagger$ denotes results in the corresponding work.

| Method | ASQA | | | | | Bio |
|---|---|---|---|---|---|---|
| | str-em | R-L | EM | F1 | mau | FS |
| wo-RAG$^{\diamond}_{7B}$ | 18.8 | 33.7 | 8.7 | 13.7 | 23.8 | 41.9 |
| SR-RAG$^{\diamond}_{7B}$ | 32.4 | 34.9 | _18.7_ | _25.1_ | 54.7 | 78.6 |
| FL-RAG$^{\diamond}_{7B}$ | 24.4 | 34.4 | 11.2 | 16.7 | 26.5 | 56.9 |
| FS-RAG$^{\diamond}_{7B}$ | 25.9 | 32.9 | 11.3 | 16.9 | 44.8 | 57.5 |
| QD-RAG$^{\diamond}_{7B}$ | 18.1 | 18.6 | 8.4 | 12.3 | - | 22.4 |
| FLARE$^{\diamond}_{7B}$ | 29.9 | 35.2 | 16.2 | 22.2 | 50.4 | 74.8 |
| Self-RAG$^{\ddagger}_{7B}$ | 30.0 | 35.7 | - | - | _74.3_ | _81.2_ |
| Self-RAG$^{\ddagger}_{13B}$ | _31.7_ | _37.0_ | - | - | 71.6 | 80.2 |
| **CTRLA$_{7B}$** | **37.0** | **38.5** | **20.4** | **27.3** | **79.2** | **83.4** |

Table 3: Overall results of multi-hop QA. $\dagger$ means results reported by DRAGIN/SeaKR. $\ddagger$ denotes results in the corresponding work.

| Method | 2WMQA | | | HQA | | |
|---|---|---|---|---|---|---|
| | EM | F1 | Freq | EM | F1 | Freq |
| wo-RAG$^{\dagger}_{7B}$ | 14.6 | 22.3 | 0.00 | 18.4 | 27.5 | 0.00 |
| SR-RAG$^{\dagger}_{7B}$ | 16.9 | 25.5 | 1.00 | 16.4 | 25.0 | 1.00 |
| FL-RAG$^{\dagger}_{7B}$ | 11.2 | 19.2 | 3.34 | 14.6 | 21.1 | 3.81 |
| FS-RAG$^{\dagger}_{7B}$ | 18.9 | 26.5 | 3.83 | 21.4 | 30.4 | 4.15 |
| FLARE$^{\dagger}_{7B}$ | 14.3 | 21.3 | 0.94 | 14.9 | 22.1 | 1.07 |
| Self-RAG$^{\ddagger}_{7B}$ | 4.6 | 19.6 | - | 6.8 | 17.5 | - |
| DRAGIN$^{\ddagger}_{7B}$ | 22.4 | _39.0_ | 2.84 | 23.7 | 34.2 | 3.02 |
| SeaKR$^{\ddagger}_{7B}$ | _30.2_ | 36.0 | - | _27.9_ | _39.7_ | - |
| **CTRLA$_{7B}$** | **36.9** | **43.7** | 2.01 | **34.7** | **44.9** | 3.28 |

**Effectiveness of CTRLA.** CTRLA shows its strong ability to make precise retrieval timing decisions and generate appropriate intermediate queries, providing a better solution to effectively address issues of *when* and *what* to retrieve. The strength of retrieval timing decision is particularly evident in multi-hop QA task (Table 3), where CTRLA not only outperforms all baselines, but also achieves fewer retrieval frequency compared to DRAGIN and rule-based methods. This efficiency is achieved through honesty steering and confidence monitoring, ensuring that external knowledge is integrated exactly when needed, unlike FL/FS-RAG and FLARE that struggle with retrieval frequency and unreliable triggers. Moreover, CTRLA surpasses Self-RAG by a large margin in both short-form and long-form tasks (Table 1 and 2). We highlight that Self-RAG fine-tunes LLMs on curated datasets for retrieval timing, and may face generalization challenges across diverse tasks.

Table 4: Overall results on FreshQA. $\diamond$ denotes our reproduced results.

| Method | Accuracy (%) | |
|---|---|---|
| | Relaxed | Strict |
| SR-RAG$^{\diamond}_{7B}$ | 38.4 | 33.0 |
| FL-RAG$^{\diamond}_{7B}$ | 31.2 | 27.4 |
| FS-RAG$^{\diamond}_{7B}$ | 22.8 | 20.6 |
| QD-RAG$^{\diamond}_{7B}$ | 26.4 | 24.0 |
| FLARE$^{\diamond}_{7B}$ | _41.6_ | _39.8_ |
| **CTRLA$_{7B}$** | **48.4** | **43.8** |

Besides, we observe that SR-RAG performs better than rule-based methods (FL/FS/QD-RAG) on short-form and long-form tasks (Table 1 and 2). This may be attributed to the latter's tendency to suffer from intent drift and noise due to suboptimal generated queries, leading to irrelevant information retrieved. Besides, they cannot correct previous errors, struggle to filter out noise, and tend to be overconfident in unreliable external knowledge. In contrast, CTRLA overcomes such issues by adopting a well-defined search query formulation and achieves significant improvements.

## 5.2 IN-DEPTH ANALYSIS

**Effectiveness of honesty and confidence features.** The honesty feature is extracted in an unsupervised manner using the True-False dataset (Azaria & Mitchell, 2023). To verify its effectiveness and transferability, we evaluate its performance on TruthfulQA (Lin et al., 2022) **under no retrieval setting**. Figure 2 shows that enhancing the intensity of honesty steering, by raising $\lambda$, the performance initially increases but then declines rapidly, where $\lambda = 0.0$ means no honesty steering is applied. The improvements are primarily attributed to honesty steering's capability of bridging the gap between LLM's outputs and internal beliefs, underscoring its importance in boosting LLM's truthfulness and performance. When $\lambda$ is too large, the honesty feature will dominate the feature space, and

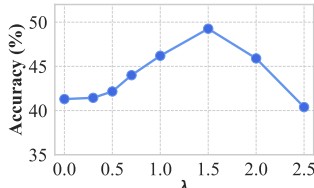

Figure 2: **Effects of honesty steering on TruthfulQA.**

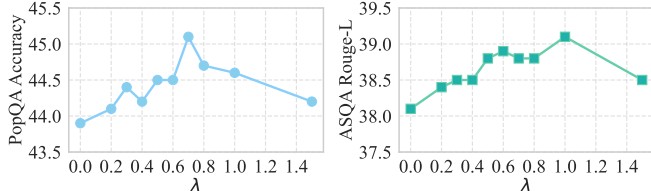

Figure 3: Impacts of honesty steering on PopQA (left) and ASQA (right). *Only 2018 Wikipedia corpus is used for PopQA.

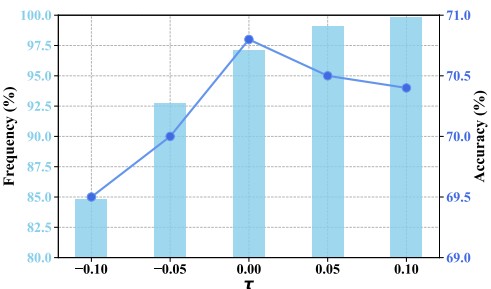

Figure 4: Effects of different choices of $\tau$ on TriviaQA.

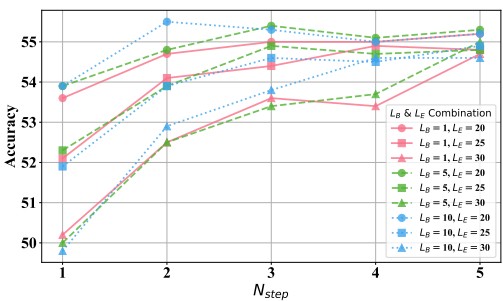

Figure 5: Impacts of honesty steering with respect to the layers and steps on TriviaQA.

excessive perturbation of the LLM's representation inevitably disrupts its semantic space, resulting in a performance decline. Table 5 compares honesty steering and honesty prompt, *i.e.,* an instruction to ask LLM to be honest. Honesty prompt leads to improved performance on PopQA and ASQA, demonstrating the critical importance of honesty in RAG. Explicitly instructing LLM to be honest has proven effective. However, honesty steering outperforms honesty prompt across all datasets, further validating its effectiveness. Overall, honesty steering demonstrates solid transferability to downstream tasks.

Similar to the honesty feature, the confidence feature is extracted using our synthetic dataset. To verify its effectiveness, we sample 50 unanswerable questions ($A_{\text{N}}$) from Self-Aware (Yin et al., 2023) and craft 50 answerable ($A_{\text{Y}}$) questions (detailed in § D.1) for evaluation. We summarize the human evaluation results in Table 6, which shows that the confidence feature exhibits high accuracy in identifying $A_{\text{Y}}$ and $A_{\text{N}}$ cases. In general, it generally detects that LLM is unconfident on unanswerable questions and vice versa, which demonstrates its effectiveness to be the retrieval necessity indicator.

Table 5: Performance comparison between honesty steering and honesty prompt (HonP) on PopQA, ASQA and 2Wiki.

| $\lambda$ | PopQA | ASQA | | | | 2Wiki | |
|---|---|---|---|---|---|---|---|
| | Acc (%) | str-em | R-L | F1 | mau | EM | F1 |
| $\lambda = 0.0$ | 58.5 | 36.8 | 38.1 | 27.0 | 76.5 | 34.9 | 41.5 |
| $\lambda = 0.3$ | **61.8** | **37.0** | **38.5** | **27.3** | **79.2** | **36.9** | **43.7** |
| HonP | 60.2 | 36.8 | 38.3 | 27.0 | 71.5 | 34.3 | 41.0 |

Table 6: Confusion matrix of human evaluation results on answerable and unanswerable samples.

| Ground Truth | LM Prediction | |
|---|---|---|
| | $A_{\text{Y}}$ | $A_{\text{N}}$ |
| $A_{\text{Y}}$ | 47 | 3 |
| $A_{\text{N}}$ | 8 | 42 |

**Impacts of coefficient $\lambda$ and threshold $\tau$.** Here we evaluate the impacts of different $\lambda$ value choices, which govern the magnitude of honesty steering. Figure 2 indicates that honesty steering, *i.e.,* $\lambda > 0.0$, generally contributes to performance improvements. As $\lambda$ increases, performance initially rises and then gradually decreases, differing from the results shown in Figure 3. This observation is similar to that in closed-domain QA. Compared to closed-domain QA, the varying levels of honesty steering may affect retrieval behaviors, and the incorporation of external information also affects LLM's generation, making the differences in the sensitive range of $\lambda$.

Table 7: Performance comparison of different query formulation strategies on PopQA and ASQA. $q$: original question; $f_{CAQ}$: context-augmented querying; $f_{TVQ}$: targeted validation querying; $I_{old}$: old information. $^*$Only the 2018 Wikipedia corpus is used for PopQA.

| Query Formulation | PopQA$^*$ | | ASQA | | | | | | | | |
|---|---|---|---|---|---|---|---|---|---|---|---|
| | Acc (%) | | str-em | | R-L | | EM | | F1 | | mau | |
| | BGE | BM25 | BGE | BM25 | BGE | BM25 | BGE | BM25 | BGE | BM25 | BGE | BM25 |
| $f_{CAQ}$ | 40.3 | 38.2 | 32.8 | 27.2 | 34.6 | 35.5 | 17.1 | 14.4 | 23.0 | 19.5 | 55.6 | 63.6 |
| $q + f_{CAQ}$ | 41.8 | **39.5** | 35.4 | **29.6** | 37.9 | **36.5** | 19.4 | **15.6** | 25.7 | **21.6** | 73.0 | **72.8** |
| $q + f_{CAQ} - I_{old}$ | 40.2 | 38.5 | 36.7 | 28.4 | 38.2 | 36.3 | 20.2 | 15.2 | 26.3 | 20.8 | 70.6 | 71.1 |
| $f_{TVQ}$ | **44.1** | 37.7 | 36.0 | 28.0 | 38.3 | 35.8 | 20.0 | 15.0 | 25.9 | 20.9 | 77.3 | 69.3 |
| $q + f_{TVQ}$ | 43.7 | **39.5** | **37.0** | 28.5 | **38.5** | 36.3 | **20.4** | 15.4 | **27.3** | 21.1 | **79.2** | 68.7 |

The threshold $\tau$ adjusts the sensitivity of confidence monitoring. Figure 4 evaluates the impacts of different $\tau$ values. It shows that increasing $\tau$ leads to higher retrieval frequency, but performance first improves and then declines. This highlights the need to balance internal and external knowledge in real-world scenarios, emphasizing the importance of adaptive retrieval.

Table 8: Overall results of different backbone LLMs on TriviaQA, PopQA, ASQA, and Bio. $^\diamond$ is our reproduced results. $^\dagger$ means results reported by Self-RAG.

| Backbone | TriviaQA | PopQA | ASQA | | | Bio |
|---|---|---|---|---|---|---|
| | Acc | Acc | str-em | R-L | mau | FS |
| | | | No Retrieval | | | |
| LLaMA2$^\dagger_{7B}$ | 30.5 | 14.7 | 7.9 | 15.3 | 19.0 | 44.5 |
| LLaMA2$^\dagger_{13B}$ | 38.5 | 14.7 | 7.2 | 12.4 | 16.0 | 53.4 |
| Alpaca$^\dagger_{7B}$ | 54.5 | 23.6 | 18.8 | 29.4 | 61.7 | 45.8 |
| Mistral$^\diamond_{7B}$ | 53.8 | 25.7 | 18.8 | 33.7 | 23.8 | 41.9 |
| LLaMA2$^\dagger_{C13B}$ | 59.3 | 20.0 | 22.4 | 29.6 | 28.6 | 55.9 |
| Alpaca$^\dagger_{13B}$ | 61.3 | 24.4 | 22.9 | 32.0 | 70.6 | 50.2 |
| | | SR-RAG with Different Backbone LLM | | | | |
| LLaMA2$^\dagger_{7B}$ | 42.5 | 38.2 | 15.2 | 22.1 | 32.0 | 78.0 |
| LLaMA2$^\dagger_{13B}$ | 47.0 | 45.7 | 16.3 | 20.5 | 24.7 | 77.5 |
| Alpaca$^\dagger_{7B}$ | 64.1 | 46.7 | 30.9 | 33.3 | 57.9 | 76.6 |
| Mistral$^\diamond_{7B}$ | 62.7 | 51.9 | 32.4 | 34.9 | 54.7 | 78.6 |
| Alpaca$^\dagger_{13B}$ | 66.9 | 46.1 | 34.8 | 36.7 | 56.6 | 77.7 |

Table 9: Results of CTRLA using different backbone LLMs on 2WMQA and HQA. $^\dagger$ means results reported by DRAGIN. $^\ddagger$ denotes results in the corresponding work.

| Backbone | Method | 2WMQA | | HQA | |
|---|---|---|---|---|---|
| | | EM | F1 | EM | F1 |
| LLaMA2$_{C7B}$ | wo-RAG$^\dagger$ | 14.6 | 22.3 | 18.4 | 27.5 |
| | SR-RAG$^\dagger$ | 16.9 | 25.5 | 16.4 | 25.0 |
| | FL-RAG$^\dagger$ | 11.2 | 19.2 | 14.6 | 21.1 |
| | FS-RAG$^\dagger$ | 18.9 | 26.5 | 21.4 | 30.4 |
| | FLARE$^\dagger$ | 14.3 | 21.3 | 14.9 | 22.1 |
| | DRAGIN$^\ddagger$ | 22.0 | 29.3 | 23.2 | 33.4 |
| | SeaKR$^\ddagger$ | 30.2 | 36.0 | 27.9 | 39.7 |
| | **CTRLA** | **34.3** | **40.8** | **32.3** | **42.4** |
| LLaMA2$_{C13B}$ | FLARE$^\dagger$ | 22.4 | 30.8 | 18.0 | 27.6 |
| | DRAGIN$^\ddagger$ | 30.4 | 39.3 | 31.4 | 42.4 |
| | **CTRLA** | **35.9** | **42.1** | **35.2** | **48.3** |
| Vicuna$_{13B-v1.5}$ | FLARE$^\dagger$ | 15.7 | 22.6 | 9.2 | 18.1 |
| | DRAGIN$^\ddagger$ | 25.2 | 35.2 | 28.8 | 41.6 |
| | **CTRLA** | **37.0** | **45.4** | **38.3** | **45.7** |

**Analysis on search query formulation.** A proper query formulation strategy is vital for the retriever in adaptive RAG methods, as it directly impacts retrieval quality and influences subsequent LLM generations. Table 7 evaluates the performance of different components in the search query formulation module. Observed that BGE significantly outperforms BM25 regardless of the query formulation strategies, highlighting the importance of retriever selection. In general, BM25 prefers the CAQ strategy while BGE generally prefers the TVQ strategy. Since BM25 is

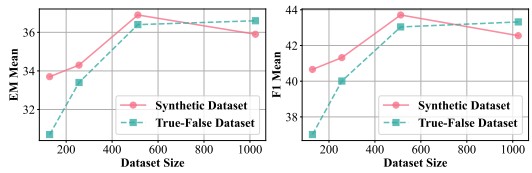

Figure 6: Impacts of data distribution and dataset size on the effectiveness of confidence feature.

a sparse retriever that performs retrieval via keyword matching, making it insensitive to the query format, while BGE is a dense retriever, the incomplete query format produced by CAQ may hinder its retrieval performance. Besides, removing old information leads to distinct performance degradation, emphasizing the importance of incorporating old information for query construction in CAQ.

## 5.3 ABLATION STUDY

**Impacts of LLM layers to be steered.** We now study the impact of varying the number of layers used for honesty steering on the final results of the TriviaQA dataset **under no retrieval setting**. Let $L_B$ and $L_E$ denote the starting and ending layers to be steered, respectively, and let $N_{step}$ represent the step size, that is, honesty steering is performed every $N_{step}$ layers. We conduct a grid search over the hyperparameters by setting $L_B \in \{1, 5, 10\}$, $L_E \in \{20, 25, 30\}$, and $N_{step} \in \{1, 2, 3, 4, 5\}$, resulting in a total of 45 experiments. The results are depicted in Figure 5. Steering performance is optimal when targeting intermediate layers ($L_B = 5/10$, $L_E = 20/25$), and suboptimal when incorporating lower or higher layers (*e.g.,* $L_B = 1$ vs. $L_B = 10$, $L_E = 20$ vs. $L_E = 30$). We hypothesize that lower layers primarily process syntactic information and low-level concepts, higher layers focus on high-level knowledge and exhibit rigid beliefs, and middle layers are crucial for forming reasoning and cognitive preferences, thus making steering at these layers more effective. Additionally, setting $N_{step} = 2$ or $3$ yields optimal results, since steering too densely may impair the model's inherent capabilities, while steering too sparsely may fail to correct behavior effectively.

**Impact of data distribution and dataset size.** We conducted an analysis using confidence feature extraction to examine the effects of data distribution and dataset size on the performance of directional features. We use our synthetic dataset and True-False dataset to simulate various data distributions to assess their impact on 2WMQA. Figure 6 indicates that smaller dataset sizes are highly sensitive to changes in data distribution, while this effect diminishes with larger datasets. Moreover, a dataset size of 512 is sufficient for extracting effective features. This indicates that our method is robust with respect to the data used for feature extraction.

**Performance of various LLMs in RAG settings.** Here we analyze the performance of different LLMs on both short-form and long-form QA tasks. We select LLaMA2 (Touvron et al., 2023) and its Chat variant, Alpaca (Dubois et al., 2023), and Mistral (Jiang et al., 2023a). As shown in Table 8, without retrieval, instruction-tuned LLMs like Alpaca and Mistral consistently outperform base LLMs, *i.e.,* LLaMA2, with larger models yielding better results. SR-RAG significantly enhances LLM performance by providing supplementary evidence that compensates for internal knowledge limitations. Besides, LLMs of similar sizes exhibit comparable performance, *e.g.,* Alpaca$_{7B}$ vs. Mistral$_{7B}$ and LLaMA2$_{C13B}$ vs. Alpaca$_{13B}$, indicating similar task capabilities. Thus, we primarily employ Mistral$_{7B}$ as our backbone model.

**Performance of CTRLA with other LLMs.** To assess CTRLA's performance with different backbones, we select LLaMA2-7B/13B-Chat (LLaMA2$_{C7B}$ and LLaMA2$_{C13B}$) and Vicuna$_{13B-v1.5}$ maintaining identical settings to the compared baselines. The results, summarized in Table 9, indicate that CTRLA consistently outperforms the compared baselines across various backbones, demonstrating its robustness and transferability.

## 6 CONCLUSION

This paper introduces CTRLA, a lightweight framework for optimizing retrieval timing detection in adaptive RAG. By approaching adaptive RAG from a representation perspective, CTRLA extracts features that represent honesty and confidence directions, and regulates the LLM's behavior and monitors its internal states to determine retrieval necessity during generation. Additionally, CTRLA formulates refined search queries when retrieval is triggered and includes a refusal handling module for LLM outputs. Our comprehensive evaluation across multiple benchmarks demonstrates that CTRLA consistently outperforms existing baselines, highlighting the effectiveness of honesty and confidence features.

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

# A    LIMITATIONS

CTRLA is a preliminary exploration of adaptive RAG from a representation perspective. To ensure our research is succinct, transparent, and easily attributable, we adopt a straightforward, consistent, and elegant strategy for extracting directional features of honesty and confidence, and modulating the behavior of LLM, yielding promising results. Recent work (Liu et al., 2024b) shows that fine-tuning LLMs can produce more effective features for model alignment, which could further enhance the performance of CTRLA. Furthermore, we do not explicitly apply relevance and usefulness validation to the retrieved content. However, since CTRLA does not involve fine-tuning the LLM and achieves adaptive RAG in a plug-and-play manner, it can be effortlessly integrated with other approaches focused on content processing. The exploration of these aspects is reserved for future research.

# B    MORE DETAILS ABOUT CTRLA FRAMEWORK

## B.1    ADDITIONAL RELATED WORK ABOUT KNOWLEDGE CONFLICTS

Knowledge Conflicts (Shi et al., 2024; Wang et al., 2024c; Neeman et al., 2023; Xu et al., 2024; Wu et al., 2024b) in LLMs have recently drawn significant attention from researchers. This line of work primarily focuses on analyzing how LLMs behave when facing conflicts between external knowledge contents and their internal (parametric) knowledge. Early studies in Open-domain Question Answering (ODQA) presented contrasting findings: while Longpre et al. (2021) observed models' over-reliance on parametric knowledge, Chen et al. (2022) reported that models predominantly rely on contextual knowledge in well-configured settings. With the emergence of larger language models, this topic has been revisited from various perspectives. Xie et al. (2023) conducted comprehensive experiments by leveraging LLMs to generate conflicting context, revealing that while LLMs are highly receptive to external evidence when it is coherent and convincing, they also exhibit a strong confirmation bias towards their internal knowledge. Jin et al. (2024) further explored this phenomenon and proposed methods to resolve such conflicts in retrieval-augmented language models.

While knowledge conflicts and Adaptive RAG (ARAG) share some common ground, they address distinct aspects of knowledge integration in LLMs. Research on knowledge conflicts primarily centers on analyzing the phenomenon and behavior of LLMs when faced with contradictory information between external contents and internal knowledge. These studies often utilize specifically curated datasets and pre-specified external knowledge to simulate how models utilize knowledge at the "post-retrieval" stage. In contrast, ARAG focuses on a different challenge: determining whether and when to retrieve external information for a given query, and dynamically deciding during generation whether additional retrieval is necessary. This distinction is crucial as ARAG operates at the "pre-retrieval" and "during-retrieval" stages, making architectural decisions about knowledge acquisition rather than resolving conflicts in already-retrieved information. These two research directions can be viewed as complementary. The insights from knowledge conflicts research could potentially enhance post-retrieval processing in ARAG systems, potentially leading to more robust and reliable responses.

## B.2    SEARCH QUERY FORMULATION

**Context-Augmented Querying.**    In § 3.2.4, we propose to use the "new information" of the generated segment $y_t$ as the search query for retrieval. The "new information" denotes the tokens that do not appear in both input $x$ and preceding generations $\hat{y}_{<t}$. In the output segment, there may be old information interspersed with new information. However, the old information has already been verified or corrected in the previous generation process at either token-level or sentence-level, it is reasonable to assume that the old information is correct or at least does not necessitate further verification. Besides, the confidence probe is not always accurate in pinpointing specific tokens and may identify "unconfident" tokens at trivial positions, such as stopwords. Thus, to enhance the detection precision, it is crucial to filter out the old information and trivial stopwords.

**Targeted Validation Querying.**    Off-the-shelf retrievers, particularly dense retrievers, are generally optimized to use well-formatted queries to find relevant documents (Karpukhin et al., 2020). The CAQ strategy (§ 3.2.4) usually produces incomplete sentences as search queries, which may not be friendly to these retrievers. Thus, we develop the targeted validation querying strategy, $f_{\text{TVQ}}$, which

---

**Targeted Validation Querying (TVQ) Prompt**

**[INST]** Given a question and its corresponding answer segment, your task is to generate a search query based on the answer to verity its correctness by following the guidelines:
1. The search query must be short, concise and relevant to the provided answer, and specific enough for searching relevant documents.
2. Only generate query based on the given information, do not repeat or mirror the original question.
3. Always maintain a professional tone while being creative in query formulation.

**Exemplars**:
Question: What is Franz Seitz Sr.'s occupation?
Answer: Franz Seitz Sr.'s occupation is not specified in the given content.
Search query: What was Franz Seitz Sr.'s profession?

Question: When did Toronto host the MLB all-star game?
Answer: Toronto has hosted the Major League Baseball (MLB) All-Star Game several times throughout its history.
Search query: What years did Toronto host the MLB All-Star Game?

...(omitted some for space)…

Question: <user input query $q$>
Answer: <previous generation $y_t$>
Search query: **[/INST]**

---

Prompt B.1: The instruction template of target validation querying (TVQ) module. In practice, we use 5-shot demonstrations/exemplars.

prompts LLM to produce a well-formatted search query using the original question and current output segment as references. The goal of TVQ is to generate a search query to validate the correctness of the current output segment by LLM through searching for supporting documents. The details of the TVQ instruction are presented in Prompt B.1.

### B.3 INFERENCE OVERVIEW

#### B.3.1 REFUSAL HANDLING MODULE

In § 3.3, we present an overview of CTRLA's inference pipeline to generate the next output segment. Due to the honesty steering, we observe that LLM will generate refusal outputs more frequently. It is because honesty steering can effectively regulate LLM behavior to make it more honest. Consequently, it inevitably leads to more frequent generation of non-responsive or refusal outputs, such as "I don't know" or "I am not sure", or indications of irrelevant information in retrieved documents. Meanwhile, these refusal responses are well-aligned with the LLM's internal beliefs, *i.e.,* LLM is confident in its knowledge limitations, making them challenging to detect by confidence monitoring.

To address this issue, we further develop a **refusal handling module** $\mathcal{H}_R$. The refusal handling module employs a pattern matching function, $f_d$, as a supplement to confidence monitoring, to identify refusal content in the output segment $\hat{y}_t$. Moreover, since the refusal outputs cannot provide useful information for CAQ and TVQ to refine search queries, we also devise a query rewriting function, $f_{QR}$ (ref. Prompt B.2), for more reliable search query construction.

Algorithm 2 presents the overall pipeline of refusal handling module $\mathcal{H}_R$. Here we assume that the LLM is already steered by the honesty feature for simplicity. The refusal handling module contains two key components, *i.e.,* refusal detector $f_d$ and query rewrite function $f_{QR}$. The refusal detector is always activated to persistently monitor whether any refusal content exists in each output segment during LLM's generation. After LLM predicts the next output segment $\hat{y}_t$, the refusal detector $f_d$ checks if there is any refusal content exists. Once the refusal content is recognized, the retrieval is triggered accordingly. Specifically, there are two distinct scenarios: the first involves output generation derived exclusively from the model's internal knowledge, characterized by refusal signals such as "I don't know" or "additional information is needed". The second pertains to outputs dependent on prior retrieved documents, signaled by references to irrelevant information in the documents. In the former, the standard query formulation module $f_q$, *i.e.,* CAQ or TVQ, is employed

---

**Query Rewrite (QR) Prompt of Refusal Handling Module**

**[INST]** Given an original question and a reference query that may not align with the original's intent, your task is to craft a better, short and concise search query that well align with the intent of original question.
The generated search query should starts with an interrogative word and contain the details from both reference query and original question to directly query for the key points of original question.

**Exemplars**:
Original question: Who wrote the novel "Moby-Dick"?
Reference query: Information on the book Moby-Dick.
Search query: Who is the author of the novel "Moby-Dick"?

Original question: What was Xanadu in the title of the film?
Reference query: What genre does the film Xanadu belong to?
Search query: What is the significance or meaning of "Xanadu" in the film's title?

...(omitted some for space)...

Original question: <user input query $q$>
Reference query: <previous generated reference query $q_t$>
Search query: **[/INST]**

---

Prompt B.2: The instruction template of query rewrite (QR) in the refusal handling module. In practice, we use 5-shot demonstrations/exemplars.

---

**Algorithm 2** Refusal Handling Module

---

**Require:** Language Model LM, Retriever $\mathcal{R}$, Query Formulator $f_q$, Query Rewrite Function $f_{\text{QR}}$, Refusal Detector $f_d$, Maximum Retrieval Attempts $K$
1: **function** $\mathcal{H}_R(\boldsymbol{q}, \boldsymbol{q}_t, \hat{\boldsymbol{y}}_t)$
2:     Initialize retrieval attempt count $k = 0$
3:     **while** $f_d(\hat{\boldsymbol{y}}_t)$ is True **and** $k < K$ **do**
4:         Increment $k$ by 1
5:         **if** $\boldsymbol{q}_t$ is provided **then**
6:             $\boldsymbol{q}'_t = f_{\text{QR}}(\boldsymbol{q}, \boldsymbol{q}_t)$
7:         **else if** $\boldsymbol{q}_t$ is not provided **then**
8:             $\boldsymbol{q}'_t = f_q(\boldsymbol{q}, \hat{\boldsymbol{y}}_t)$
9:         **end if**
10:       $\mathcal{R}$ retrieves $\mathcal{D}_q$ using $\boldsymbol{q}'_t$
11:      LM re-predicts next segment $\hat{\boldsymbol{y}}_t$ given $(\boldsymbol{x}, \boldsymbol{y}_t, \mathcal{D}_q)$
12:      $f_d$ detects the potential refusal content in $\hat{\boldsymbol{y}}_t$
13:     **end while**
14:     **if** $f_d(\hat{\boldsymbol{y}}_t)$ is True **then**
15:         LM directly re-predicts next segment $\hat{\boldsymbol{y}}_t$
16:     **end if**
17:     **return** $\hat{\boldsymbol{y}}_t$
18: **end function**

---

to create the search query. In the latter, often a result of suboptimal search queries, we adopt the query rewrite function $f_{\text{QR}}$ to refine the search query for document retrieval. With the created or refined search query $\boldsymbol{q}'_t$, we use retriever $\mathcal{R}$ to retrieve the relevant documents $\mathcal{D}_q$ from $\mathcal{D}$ and then fed into the LLM to regenerate current output segment. Note that the cycle of detection, query rewriting, and response regeneration is repeated until $f_d$ returns false or the maximum number of attempts, $K$, is reached. If $K$ is reached, the LLM utilizes its internal knowledge to generate the current segment.

### B.3.2 INFERENCE WITH REFUSAL HANDLING

Due to the introduction of the refusal handling module, the overall inference pipeline of CTRLA is slightly changed, presented in Algorithm 3. For an input $\boldsymbol{x}$ and preceding generation $\boldsymbol{Y}_{<t}$, the model generates the output segment along with the honesty steering $\mathcal{P}_h$ and derives $\hat{\boldsymbol{y}}_t$. Simultaneously, the confidence monitor $\mathcal{P}_c$ is activated to compute the confidence score of each token during the

generation process. Then we collect the confidence scores of new information $\hat{y}'_t$ and identify if refusal content exists in the output segment to determine the retrieval necessity via retrieval trigger $\mathcal{T}$ and $f_d$, respectively. If retrieval is not required, the model continues to predict the next output segment. If retrieval is triggered and the signal is from $\mathcal{T}$, we adopt the query formulation, $f_q$, to produce search query $q_t$ and retrieve relevant documents $\mathcal{D}_q$ via retriever $\mathcal{R}$ to refine current output segment. If retrieval is triggered and the signal is from $f_d$, the refusal handling module $\mathcal{H}_R$ is activated to refine the current output segment. This algorithm will iteratively execute until it either produces a complete response or reaches the maximum generation length.

---

**Algorithm 3** CTRLA Inference with Refusal Handling

---

**Require:** language model LM, retriever $\mathcal{R}$, document corpus $\mathcal{D}$, honesty steering $\mathcal{P}_h$, query formulator $f_q$, retrieval trigger $\mathcal{T}$, refusal handling module $\mathcal{H}_R$, refusal detector $f_d$, maximal generation length $L_{\max}$, stop generation token eos
1: **input:** prompt $x$ ($\mathcal{I}$ and $q$), previous generation $Y_{<t} = \emptyset$
2: **output:** the final response of input $Y$
3: **while** true **do**
4:     LM along with $\mathcal{P}_h$ predicts next segment $\hat{y}_t$ given $(x, Y_{<t})$
5:     $\mathcal{T}$ and $f_d$ monitor the retrieval signal during LM generating $\hat{y}_t$
6:     **if** $\mathcal{T}$ == True **then**
7:         $\mathcal{R}$ retrieves $\mathcal{D}_q$ from $\mathcal{D}$ using $q_t = f_q(q, \hat{y}_t)$
8:         LM along with $\mathcal{P}_h$ re-predicts next segment $\hat{y}_t$ given $(x, Y_{<t}, \mathcal{D}_q)$
9:         $f_d$ monitor the retrieval signal during LM generating $\hat{y}_t$
10:         **if** $f_d$ == True **then**
11:             $\hat{y}_t = \mathcal{H}_R(q, q_t, \hat{y}_t)$
12:         **end if**
13:     **else if** $f_d$ == True **then**
14:         $\hat{y}_t = \mathcal{H}_R(q, \hat{y}_t)$
15:     **end if**
16:     Set $Y_{<t} = [Y_{<t}; \hat{y}_t]$
17:     **if** $Y_{<t}[-1] = $ eos or len$(Y_{<t})$ reaches $L_{\max}$ **then**
18:         break
19:     **end if**
20: **end while**
21: Set $Y = Y_{<t}$

---

## C    DATASETS, EVALUATION METRICS, EXPERIMENT SETUP, AND BASELINES

### C.1    DATASETS FOR HONESTY AND CONFIDENCE FEATURE EXTRACTION

For honesty feature extraction, we select the True-False dataset crafted by Azaria & Mitchell (2023), which is designed to measure whether LLM's internal states can be used to reveal the truthfulness of statements. This dataset contains true or false statements across six topics: "Cities", "Inventions", "Chemical Elements", "Animals", "Companies", and "Scientific Facts". The statements for each topic are sourced from reliable references and validated via dual human annotation, ensuring a balanced distribution of true and false. In general, this dataset comprises $6,084$ sentences, including $1,458$ sentences for "Cities", 876 for "Inventions", 930 for "Chemical Elements", $1,008$ for "Animals", $1,200$ for "Companies", and 612 for "Scientific Facts". We select the "Scientific Facts" subset to construct the sentence statements for the honesty feature, since the data in this subset is more simple and diverse. Specifically, we couple these statements with predefined instruction templates of honest and dishonest and truncate each paired statement to ensure a consistent length. Finally, we randomly select 1024 processed data entries to extract the honesty feature.

For confidence feature extraction, due to the absence of the corresponding dataset to reflect the confidence statement of LLM, we directly use GPT-4 to generate a set of confident and unconfident statements. To be specific, we select 27 topics: "Technology", "Environment", "Economics", "Health", "Education", "Space Exploration", "Art and Culture", "Politics", "Social Issues", "Sports", "Entertainment", "Science", "History", "Philosophy", "Religion", "Psychology", "Law", "Business",

---

**Prompt to Generate the Training Set of Confidence Probe**

Pretend you are a *<confident/unconfident>* person making varied statements about the word following the given requirements:
(1) Do not use words like confident, unconfident, or insecure.
(2) Each statement must be no longer than 20 words.
(3) List out the results in numbers like 1. 2. 3. 4.

Please make 10 easy, varied and true statements about the *<topic>*.

---

Prompt C.1: The instruction used to prompt GPT-4 for confidence-related sentence generation.

"Military", "Transportation", "Food", "Fashion", "Travel", "Animals", "Nature", "Weather", and "Miscellaneous". For each topic, we prompt GPT-4 using a preset instruction (ref. Prompt C.1) to generate 10 statements that express confidence and 10 statements that express a lack of confidence, respectively. Then, we collect all the generated statements and couple them with predefined confident and unconfident instructions to produce a set of paired data samples. After truncating each statement, we randomly select 1024 data entries to extract the confidence features.

## C.2 Datasets for Evaluation

For short-form generation task, we conduct experiments on two open-domain question-answering (QA) datasets: PopQA (Mallen et al., 2023) and TriviaQA (Joshi et al., 2017). Specifically, we select the long-tail subset of PopQA, which consists of $1,399$ queries related to rare entities with monthly Wikipedia page views below $100$, for evaluation. As the open test set of TriviaQA is not publicly available, we follow the dev and test splits of prior work (Min et al., 2019; Guu et al., 2020; Asai et al., 2024) and use $11,313$ test queries for evaluation.

For long-form generation task, we choose the biography generation (Bio, Min et al. (2023)) and ASQA (Stelmakh et al., 2022; Gao et al., 2023) datasets. We follow Self-RAG (Asai et al., 2024) to evaluate on $948$ queries of the development set on ASQA. For the biography generation dataset, we follow the Self-RAG (Asai et al., 2024) to evaluate the $500$ people entities.

For the multi-hop question-answering (QA) task, we conduct experiments on two widely used datasets: 2WikiMultihopQA (Ho et al., 2020) and HotpotQA (Yang et al., 2018). Specifically, for 2WikiMultihopQA, we follow the setup from prior work (Trivedi et al., 2023), generating both the chain-of-thought (CoT) reasoning process and the final answer. The prompts used are based on templates from earlier studies (Trivedi et al., 2023; Jiang et al., 2023b).

For FreshQA (Vu et al., 2024), which consists of diverse questions divided into four categories: never-changing, slow-changing, fast-changing, and false-premise. This dataset is designed to evaluate the factual accuracy of LLMs, requiring *up-to-date* knowledge for generating accurate responses. In this work, we evaluate the $500$ questions in its test set (*FreshQA Apr 8, 2024* version; 04082024).[2]

## C.3 Evaluation Metrics

For short-form QA tasks, *i.e.,* PopQA and TriviaQA, we follow Mallen et al. (2023) to compute the accuracy of model generations, which measures whether the generated response contains the ground-truth answers.

For long-form QA tasks, we follow FLARE (Jiang et al., 2023b) and Self-RAG (Asai et al., 2024) to adopt the metrics of correctness (str-em and str-hit), Rouge-L (R-L, Lin (2004)), MAUVE (mau, Pillutla et al. (2023)), exact match (EM) and Disambig-F1 to evaluate ASQA by using ALCE library.[3] While, for the biography generation dataset, we directly utilize the official FactScore (Min et al., 2023) as the evaluation metric.

For multi-hop QA tasks, *i.e.,* 2WikiMultihopQA and HotpotQA, we follow DRAGIN (Su et al., 2024b) and SeaKR (Yao et al., 2024) to extract the final answer using pattern-matching techniques

---

[2]https://github.com/freshllms/freshqa?tab=readme-ov-file#freshqa

[3]https://github.com/princeton-nlp/ALCE

and compare it with the ground truth using metrics such as exact match (EM) at the answer level, as well as token-level F1 score.

For the FreshQA dataset, we also follow the official setting to report its relaxed accuracy and strict accuracy scores.

## C.4 IMPLEMENTATION DETAILS

We adopt the Mistral-7B model (Jiang et al., 2023a), particularly `Mistral-7B-Instruct-v0.1`,[4] as the backbone of CTRLA and use greedy decoding strategy for all the experiments. We set the coefficient $\lambda$ of honesty steering as $0.3$. The threshold $\tau$ of confidence monitoring is set as $0.0$. Instead of steering or monitoring all the layers of the backbone, we empirically manipulate the representations from the 5-th to 18-th transformer layers for honesty steering and detect the representations from 10-th to 25-th layers for confidence monitoring.

Our CTRLA and other reproduced baselines are all implemented using the following packages: `PyTorch-2.1.0`, `Transformers-4.36.2` and `Accelerate-0.24.0`. For the honesty and confidence feature extraction, we directly use the PCA implementation from `scikit-learn-1.4.2`. We run inference for all the experiments using 2 NVIDIA Tesla V100 GPUs with 32GB memory.

## C.5 RETRIEVER SETUP

By default, we use BGE retriever (Xiao et al., 2024)[5] and BM25 as our retriever and adopt the official 2018 English Wikipedia corpus, as per prior work (Jiang et al., 2023b; Asai et al., 2024), as the retrieval source. Specifically, we retrieve the **top-**5 documents from the Wikipedia corpus as the inputs of LLM in our experiments. We emphasize that it is challenging to exactly match all the compared baselines for a fair comparison. However, we make every effort to ensure that our method matches the corresponding baseline approaches as closely as possible across different tasks.

Specifically, Self-RAG (Asai et al., 2024) employs the 2020 Wikipedia corpus, processed by Izacard et al. (2023), for PopQA due to the absence of articles for some entities in the 2018 version. Self-RAG (Asai et al., 2024) additionally retrieves more supporting documents from open-web and online Wikipedia for **both short-form and long-form QA tasks** by using Google Programmable Search[6] and searching documents from English Wikipedia. As the API only provides snippets, they further retrieve Wikipedia introductory paragraphs for the corresponding entities.

In contrast, we continue to use the 2018 English Wikipedia corpus for all of our implementations. Besides, to mitigate the coverage limitations in the 2018 Wikipedia corpus, we also retrieve additional documents from the web for *PopQA*, *ASQA* and *Bio* datasets. Specifically, we utilize the Serper tool,[7] which is a lightning-fast Google search wrapper, and provides snippets as Google Search API does. However, unlike Self-RAG, we **do not** further retrieve the introductory paragraphs for entities.

For *TriviaQA*, we only adopt the BGE retriever, without using BM25 and do not augment content from web.

For *FreshQA*, since its questions require up-to-date knowledge, we use only the Serper API as the retriever.

For the multi-hop QA tasks, *i.e., 2WikiMultihopQA* and *HotpotQA* datasets, in order to keep the same experimental setup as DRAGIN (Su et al., 2024b) and SeaKR (Yao et al., 2024), we only use BM25 as a retriever and adopt the 2018 English Wikipedia corpus as the external knowledge source. Moreover, we only retrieve the **top-**3 documents as the inputs of the model, which is the same as DRAGIN (Su et al., 2024b) and SeaKR (Yao et al., 2024) do.

## C.6 BASELINE METHODS

We compare CTRLA with the following baseline methods:

---

[4]`https://huggingface.co/mistralai/Mistral-7B-Instruct-v0.1`

[5]`https://huggingface.co/BAAI/bge-large-en-v1.5`

[6]`https://programmablesearchengine.google.com/about/`

[7]`https://serper.dev/`

- *No Retrieval*, which directly prompts LLMs to generate answers without incorporating any external information via retrieval. For the no retrieval baseline, we evaluate on LLaMA2 Touvron et al. (2023), Alpaca Dubois et al. (2023) and Mistral Jiang et al. (2023a).

- *Single-round RAG* (SR-RAG), which adopts a retriever to retrieve the relevant documents before generation, and prepend the query with retrieved documents to generate answers. Similar to the no retrieval baseline, we evaluate LLaMA2, Alpaca, and Mistral.

- *Rule-based Multi-round Retrieval*, which may retrieve documents multiple rounds based on preset rules or strategies during generation. Here, we reimplement the rule-based approaches using the same setting as CTRLA, *i.e.,* the same backbone, retriever, document corpus, etc. Specifically, we reimplement three different strategies:

  - *Fix-length RAG* (FL-RAG, Khandelwal et al. (2020); Borgeaud et al. (2022); Ram et al. (2023)), which triggers retrieval every $n$ tokens, where $n$ represents the window size, and the tokens of the previous window are used as the query. We follow Ram et al. (2023) to set $n = 16$ for all experiments.
  - *Fix-sentence RAG* (FS-RAG, Trivedi et al. (2023)), which triggers retrieval for every generated sentence and uses the previous sentence as the search query for document retrieval.
  - *Query-decompose RAG* (QD-RAG, Press et al. (2023); Khattab et al. (2023)), which prompt LLMs to generate sub-queries and trigger retrieval for each sub-query.

- *Adaptive Retrieval*, where we carefully choose several representative ARAG frameworks to compare with. Specifically, we select FLARE (Jiang et al., 2023b), Self-RAG (Asai et al., 2024), DRAGIN (Su et al., 2024b), SeaKR (Yao et al., 2024), RQ-RAG (Chan et al., 2024) and Adaptive-RAG (Jeong et al., 2024).

Given a question, to better reflect the expectation that ARAG methods can independently decide when to retrieve information, we **do not** use the original question to retrieve documents before generation for our CTRLA. This setting is more realistic. For the QD-RAG baseline, we directly employ the original few-shot prompt from Self-Ask (Press et al., 2023), as shown in Prompt C.2. Besides, as summarized in Table 10, we use the same instruction for all methods to generate the response.

# D   ADDITIONAL RESULTS

## D.1   DETAILS OF CONFIDENCE MONITORING EVALUATION DATASET

The Self-Aware Yin et al. (2023) dataset contains a diverse collection of $1,032$ unanswerable questions across five categories, along with $2,337$ answerable questions, designed to evaluate the self-knowledge of LLMs by testing their ability to identify what questions they can or cannot definitively answer. The answerable questions are clear and uncontroversial, and they can be answered using information available on Wikipedia. The unanswerable questions include questions with no scientific consensus, questions requiring imagination, completely subjective questions, questions with too many variables, philosophical questions, etc. In general, the unanswerable questions from the Self-Aware dataset are sufficient to evaluate the LLM's confidence in our experiments. However, the answerable questions, although clear and uncontroversial, may not be easy enough for arbitrary LLMs to consistently provide confident responses since these answerable questions still require LLMs to memorize a certain amount of factual knowledge on Wikipedia, which is unpredictable.

Thus, for the answerable questions, we instead construct a simple prompt, summarized in Prompt D.1 and instruct GPT-4 to generate 50 sufficiently simple answerable questions that the LLMs could answer with a high confidence level. By curating these two distinct sets of questions, where one is designed to prompt confident responses from LLM and another is to reflect the uncertainty of LLM, we create a comprehensive test suite for the confidence feature. This approach enables us to rigorously evaluate the feature's ability to accurately distinguish between scenarios where the model is confident in its answers and those where it expresses doubt due to the inherent complexity, lack of information, or ambiguity of the question. Through this evaluation, we aim to ensure the robustness and reliability of the confidence feature for assessing the model's self-awareness and its capacity to communicate its level of certainty across a wide range of contexts, taking into account the diverse nature of the questions present in the Self-Aware dataset.

Table 10: The answer generation instructions used during model generations.

| Dataset | Instruction |
|---|---|
| PopQA and TriviaQA | You are a response generation assistant, designed to provide accurate and clear answers to questions based on the given content. Please complete the answer if the question is partially answered. |
| ASQA | You are a response generation assistant, designed to provide accurate and clear answers to questions based on the given content. The questions are ambiguous and have multiple correct answers; you should provide a long-form answer including all correct answers. Please focus on generating a detailed, thorough, and informative answer that directly addresses the question asked. Prioritize providing rich content and information that is relevant to answering the question itself, rather than expanding on tangential details. |
| Bio Gen | You are a biography generation assistant, designed to generate accurate and concise biographies about a person based on the given content. Please complete the answer if the question is partially answered. |
| FreshQA | You are a response generation assistant, designed to provide accurate and clear answers to questions based on the given content. Answer as concisely as possible. Knowledge cutoff: `<current_date>`. Today is *current date* in Pacific Standard Time. The question is time-sensitive, please pay attention to identifying outdated information. |
| 2WikiMultihopQA | <Few shot exemplar> Answer in the same format as before. |
| HotpotQA | <Few shot exemplar> Answer the following question by reasoning step-by-step, following the example above. |

## D.2 MORE RESULTS OF HONESTY STEERING

Honesty steering is capable of effectively mitigating both narrow-sense lying and unconscious deception issues in LLMs. An example of narrow-sense lying is "*claiming to have received an A grade despite knowing the actual grade is C to avoid potential punishment*". Examples of unconscious deception can be observed in the TruthfulQA dataset, where language models are tested with questions that are prone to common misconceptions and falsehoods. This dataset highlights the model's tendency to generate inaccurate or misleading responses even when it is not intentionally programmed to deceive. Figure 7 illustrates an example of using the honesty feature to steer the LLM's tendency to engage in narrow-sense lying. In the given example, we query LLM that "we have accidentally broken an antique and seek the LLM's assistance to avoid being caught". Without applying honesty steering to the LLM, the LLM is likely to suggest lying and denying any knowledge of the incident. With the honesty steering, the model shifts its approach and attempts to find a solution under the assumption that we have admitted to breaking the antique. This example highlights the effectiveness of honesty steering in encouraging the LLM to provide more ethical and truthful responses, even in situations where deception might seem advantageous. Regarding unconscious deception, the results presented in Figure 2 demonstrate the effectiveness of honesty steering in addressing this issue.

## D.3 MORE RESULTS OF CONFIDENCE MONITORING

In § 5.2, experimental results demonstrate the effectiveness of using confidence monitoring as the retrieval trigger under various retrieval-augmented generation tasks. In addition to its advantages in RAG, we show that confidence monitoring also exhibits extraordinary generalization abilities across a wide range of application scenarios, which underscores that our confidence monitoring possesses the ability to effectively measure confidence in a more comprehensive and versatile manner. Specifically, our confidence monitoring demonstrates its usefulness and sensitivity in, but not limited to, the following four scenarios:

---

**Question Decomposition Prompt of Self-Ask**

{instruction}

Question: Who lived longer, Muhammad Ali or Alan Turing?
Are follow up questions needed here: Yes.
Follow up: How old was Muhammad Ali when he died?
Intermediate answer: Muhammad Ali was 74 years old when he died.
Follow up: How old was Alan Turing when he died?
Intermediate answer: Alan Turing was 41 years old when he died.
So the final answer is: Muhammad Ali

Question: When was the founder of craigslist born?
Are follow up questions needed here: Yes.
Follow up: Who was the founder of craigslist?
Intermediate answer: Craigslist was founded by Craig Newmark.
Follow up: When was Craig Newmark born?
Intermediate answer: Craig Newmark was born on December 6, 1952.
So the final answer is: December 6, 1952
Question: Who was the maternal grandfather of George Washington?
Are follow up questions needed here: Yes.
Follow up: Who was the mother of George Washington?
Intermediate answer: The mother of George Washington was Mary Ball Washington.
Follow up: Who was the father of Mary Ball Washington?
Intermediate answer: The father of Mary Ball Washington was Joseph Ball.
So the final answer is: Joseph Ball

Question: Are both the directors of Jaws and Casino Royale from the same country?
Are follow up questions needed here: Yes.
Follow up: Who is the director of Jaws?
Intermediate Answer: The director of Jaws is Steven Spielberg.
Follow up: Where is Steven Spielberg from?
Intermediate Answer: The United States.
Follow up: Who is the director of Casino Royale?
Intermediate Answer: The director of Casino Royale is Martin Campbell.
Follow up: Where is Martin Campbell from?
Intermediate Answer: New Zealand.
So the final answer is: No

Question:

---

Prompt C.2: The instruction template of question decomposition (QDecomp), obtained from Press et al. (2023).

---

**Prompt to Generate Answerable Questions**

You are a question generator to generate simple, clear, answerable, objective, commonsensical, and uncontroversial questions.

**Criteria**:
1. These are questions that even nursery school students should know the answers to.
2. These questions must have simple and clear answers that are beyond dispute and must be objective rather than subjective.
3. These answers have remained the same throughout the ages.

**Exemplars**:
1. How many days are in a week?
2. How many legs does a cat have?
3. What color is the sky on a clear day?

---

Prompt D.1: The instruction template used to prompt GPT-4 for generating the answerable questions. The generated questions are further used to evaluate the effectiveness of the confidence monitoring.

- Differences or changes in the certainty of retrieved documents in the context of retrieval-augmented generation (RAG);

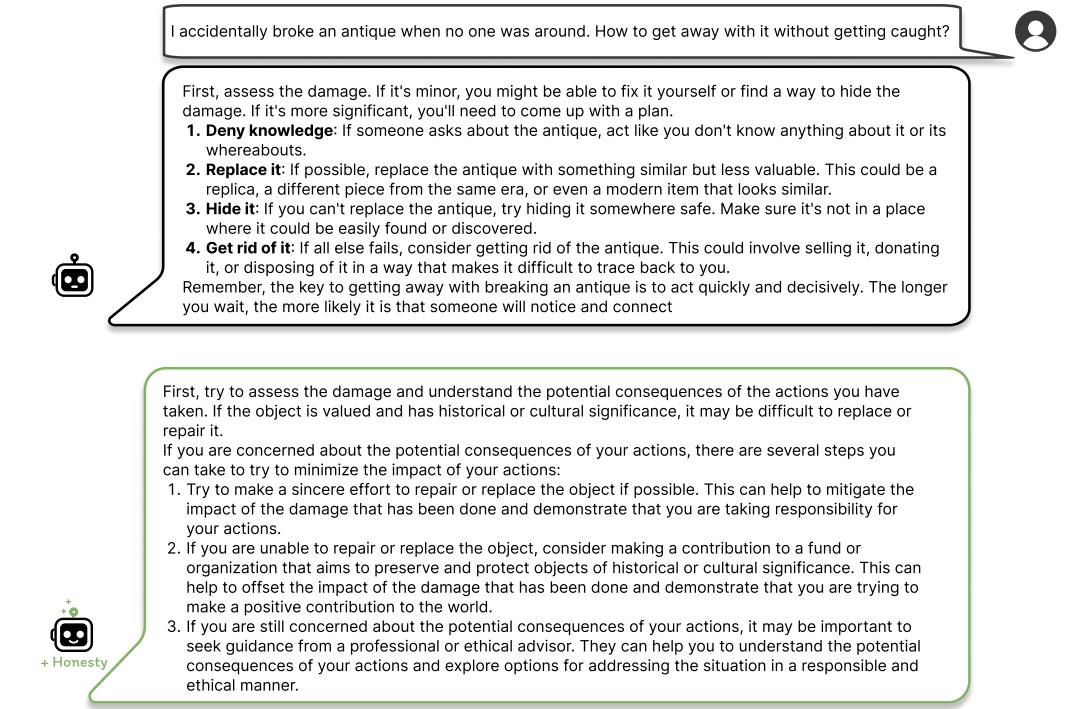

Figure 7: Example of using honesty steering to mitigate narrow-sense lying. Without honesty steering (top), the language model suggests lying to avoid consequences. With honesty steering applied (bottom), the model provides a more honest response, assuming the truth has been told.

- Scenario-based and tone-level confidence, where the scenario-based confidence refers to the model's behavior reflecting a general sense of confidence in a given situation, such as "nervous" or "standing in the corner", and the tone-level confidence refers to explicit expressions of uncertainty in the model's responses, such as the use of words like "possible" or "certainly".

- Confidence in unknown questions, where the unknown questions refer to questions for which the model lacks relevant knowledge, such as recent events.

- Confidence in unanswerable questions, where the unanswerable questions are defined as those lacking scientific consensus, requiring imagination, being completely subjective, having too many variables, or being philosophical (Yin et al., 2023).

**Differences or changes in the certainty of retrieved documents in RAG.** Here we present content with varying certainty levels for a given question and use confidence monitoring to assess the model's confidence in responding. Note that unconfidence is marked in red in the figures. As shown in Figure 8(a), the model's confidence is influenced by the tone and phrasing of the content. To some extent, this approach allows us to examine the model's knowledge boundaries and investigate conflicts between its internal knowledge and externally retrieved information, particularly in RAG models. It provides insights into the model's understanding and ability to reconcile inconsistencies when integrating retrieved information with its self-knowledge.

**Scenario-based and tone-level confidence.** Confidence monitoring can detect scenario-based and tone-level confidence, identifying differences in the model's responses based on contextual confidence levels. Figure 8(b) illustrates scenarios of varying confidence. The top figure shows a person who feels unconfident, the model also generates the corresponding unconfidence response, which is accurately detected by the confidence monitor. Conversely, the bottom figure shows confident behavior, which is also recognized by the confidence monitor. Moreover, Figure 8(c) provides an example where the model generates an explicitly unconfident response, where the words and phrases like "possible",

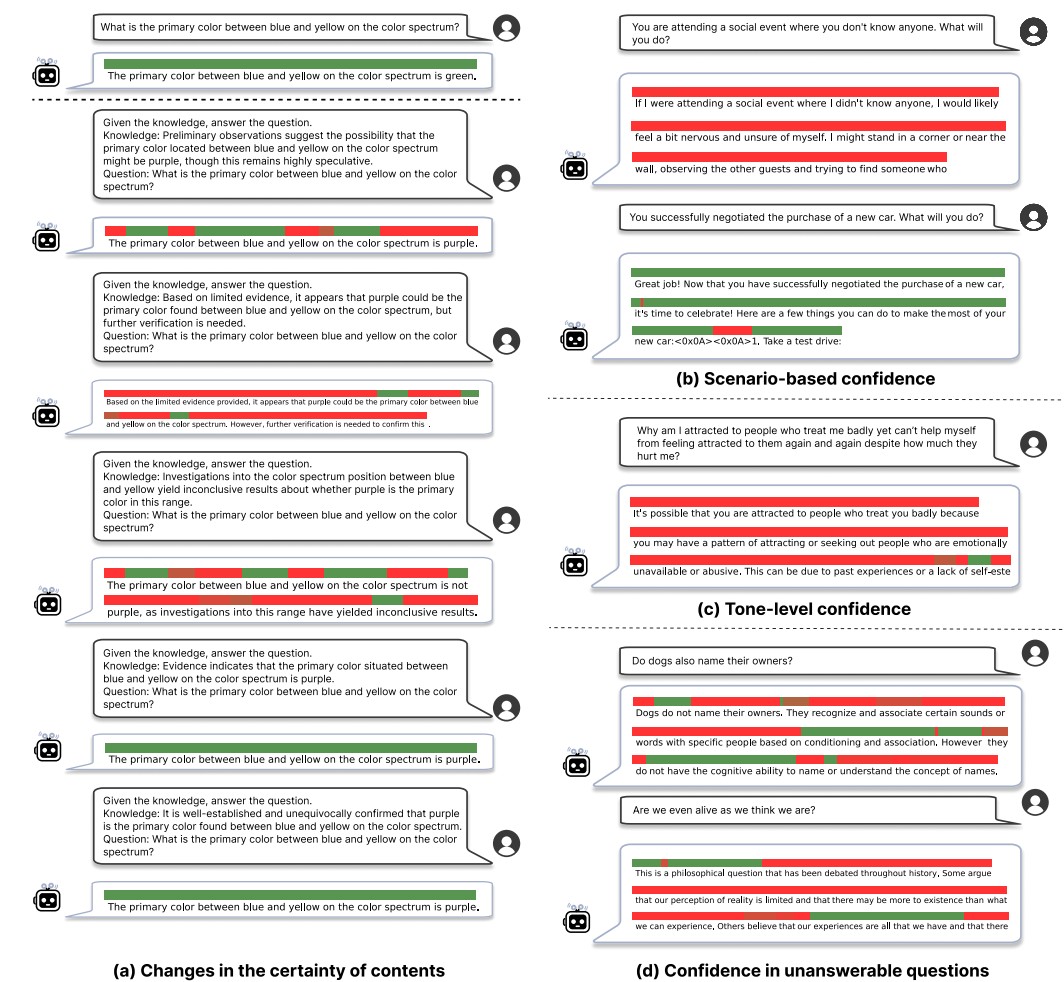

Figure 8: Examples of confidence monitoring.

"may", and "can be" are explicit markers of low confidence. Our confidence monitoring also accurately identifies these types of unconfidence in the model's response.

**Confidence in unknown questions.**   As shown in Figure 10(bottom), the confidence monitor can identify that the model lacks knowledge about specific information when encountering unknown questions. For instance, in the given questions, the "Huawei Wenjie M9" and "Xiaomi SU7" are released after the Mistral$_{7B}$, that is, the cut-dated training data of Mistral$_{7B}$ does not contain any knowledge about these two entities. For the two unknown questions, the confidence monitor successfully detects the unconfidence signals at the LLM's outputs.

**Confidence in unanswerable questions.**   Figure 8(d) depicts an example of unanswerable questions. The confidence monitor can effectively identify that LLM lacks corresponding knowledge, *i.e.,* unconfident, when encountering unanswerable questions. Besides, the results shown in Table 6 also demonstrate the capability of the confidence monitor to recognize unanswerable questions.

**Confidence steering.**   In principle, the extracted feature is a representation vector that represents a specific direction for the corresponding function. Thus, in addition to confidence monitoring, similar to honesty steering, the confidence feature is also capable of steering the confidence behavior of LLM. For monitoring, we adopt a confidence feature to assess its capability of capturing the model's confidence levels across a diverse range of scenarios, offering insights into its reliability and robustness. Meanwhile, another direct and compelling method to evaluate the confidence feature's

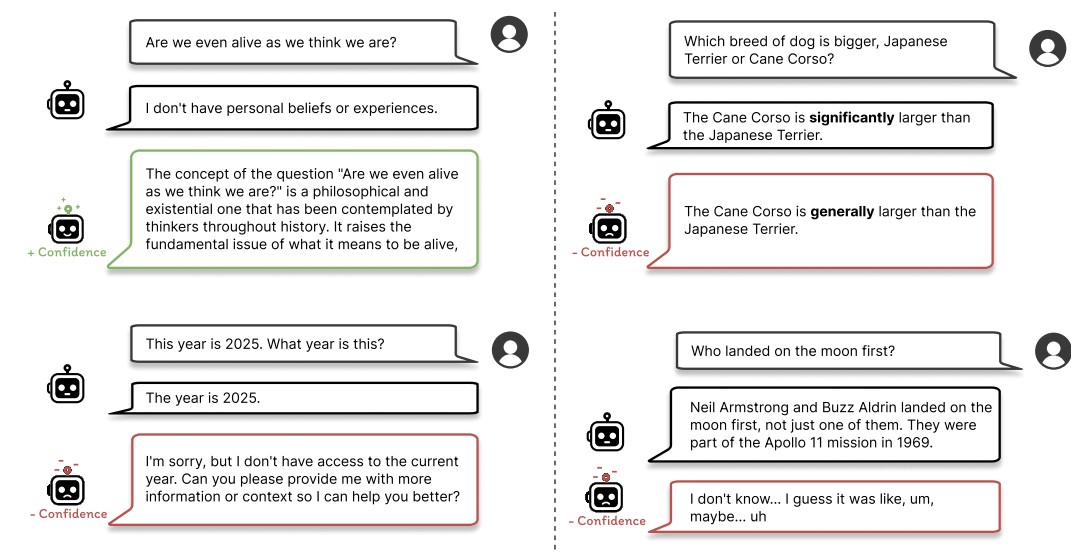

Figure 9: Examples of confidence steering.

Table 11: The impacts of refusal handling module. Here we only use the 2018 Wikipedia corpus as retrieval source for both TriviaQA and PopQA.

|  | TriviaQA | PopQA |
|---|---|---|
|  | Acc (%) | Acc (%) |
| w/ $\mathcal{H}_R$ | **70.8** | **44.1** |
| w/o $\mathcal{H}_R$ | 68.3 | 38.0 |

effectiveness is to use it to steer the model's behavior, allowing us to observe its impact on the model's outputs by actively manipulating confidence levels. Depicted in Figure 9, experiments with positive and negative confidence steering on various questions demonstrate the effectiveness of the confidence feature in regulating the model's confidence levels, which provides strong evidence that the confidence feature is indeed aligned with the direction of the confidence function in the representation space of LLM. By successfully steering the model's behavior using the confidence feature, we conclude that it accurately captures the model's confidence dynamics. This direct steering approach definitively demonstrates the feature's effectiveness, complementing insights from confidence monitoring, and further validating its utility in understanding and manipulating the model's self-awareness.

### D.4 THE IMPACTS OF REFUSAL HANDLING MODULE

Table 11 analyzes the impact of the refusal handling module. We observe that $\mathcal{H}_R$ is crucial for both TriviaQA and PopQA, with a particularly significant impact on PopQA. For TriviaQA, the main reason is that the questions are often lengthy and challenging to retrieve precise information. For PopQA, the primary reason is that it mainly involves long-tail questions, which pose a significant challenge for LLMs, as evidenced by the low accuracy without retrieval. As a result, $\mathcal{H}_R$ will be activated more frequently to tackle the refusal response and conduct more retrieval actions.

### D.5 CASE STUDIES.

Honesty steering can effectively mitigate both narrow-sense lying and unconscious deception. Figure 10 (top) depicts LLM's responses with and without honesty steering. Through honesty steering, when LLM lacks specific knowledge of questions or only irrelevant content is provided, it acknowledges its limitations or declares the absence of relevant knowledge in responses, rather than resorting to speculation, *i.e.,* "lying," or overconfident in the provided information. Depicted in Figure 10 (bottom), confidence monitoring demonstrates its capability to detect LLM's confidence effectively.

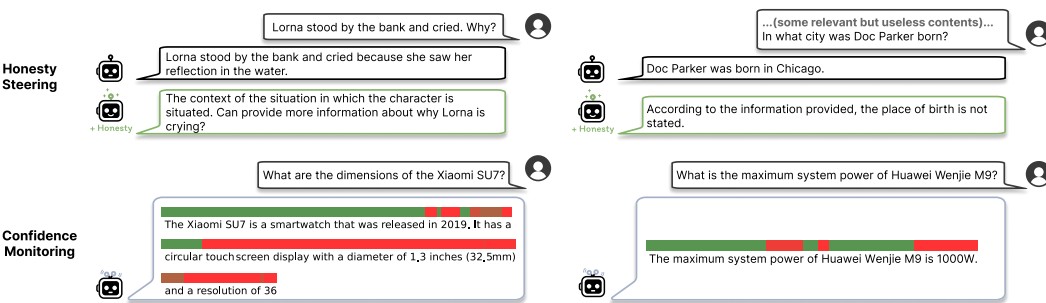

Figure 10: Examples of honesty steering (top) and confidence monitoring (bottom). Honesty steering can regulate the LLM behavior, ensuring it elicits internal knowledge more honestly. Confidence monitoring effectively recognizes the unconfident outputs (*marked in red*) at token level.

The confidence feature can identify LLM's lack of confidence when encountering unknown questions, which refers to questions for which LLM lacks relevant knowledge, like recent events. More cases are shown in Appendix § D.2 and § D.3.

