# OpenReview forum: "CtrlA: Adaptive Retrieval-Augmented Generation via Inherent Control"
_ICLR.cc/2025/Conference — Submitted to ICLR 2025_

### Official Review · Reviewer_nBtC · 2024-10-27

**Soundness:** 3
**Presentation:** 2
**Contribution:** 2
**Rating:** 5
**Confidence:** 4

**Summary:**

This paper approaches the problem of adaptive retrieval for RAG, in which “when and what to retrieve” is dynamically determined during generation. Motivated by the limitation of existing uncertainty-based approaches, this paper proposed to consider both “honesty” and “confidence” as important factors to guide adaptive retrieval. Three techniques are introduced: (1) shifting LLM representation to the honest direction, (2) projecting the representation onto a confidence feature and (3) a revised query generation technique based on the already-generated content. Experiments using several LLMs on short-form and long-form generation tasks verify the effectiveness of the proposed approach compared to existing adaptive retrieval methods.

**Strengths:**

* Although the adaptive retrieval framework has been introduced for several years, it is still an open problem for how to build strong adaptive retrieval strategies catered to modern LLM. The research question proposed by the paper is thus timely.

* The empirical evaluation is thorough. A number of LLMs are evaluated with the proposed method. The authors consider several major benchmarks across different generation formats. The empirical results can serve as good reference points for future works on adaptive retrieval.

* The proposed method is intuitive and can outperform well-known existing baselines such as FLARE and DRAGIN.

**Weaknesses:**

My major concern of this paper is that its proposal and contribution are not aligned with the key problem the paper is motivated by.

* This paper contains many moving parts, but most of them are (1) disconnected and (2) irrelevant to the key challenge of adaptive retrieval. The core motivation of this paper is that uncertainty-based adaptive retrieval is insufficient for determining when retrieval should be triggered. However, the proposed honesty steering and query formulation methods aim to generate higher quality RAG responses instead of solving the problem of adaptive retrieval. In addition, the proposed retrieval triggering criteria is still fundamentally an uncertainty-based method.

* The proposed method does not seem to solve the issue in the motivating example, where adaptive retrieval fails to trigger when the model confidently says “I don’t know”. From my understanding, the proposed method will also fail in this case.

* Algorithm 1 is incomplete. Specifically, an outer loop that calls algorithm 1 is required. The input to the outer loop is the query and the document, and the output is the final generated response, instead of a single output segment.

* More citations to the adaptive RAG literature are required in section 3. Specifically, 3.1 and 3.2.4 should cite FLARE [1], as the approaches are similar. In addition, decoding-stage feature monitoring works such as [2] and [3] should be cited in the confidence monitoring section,

[1] Active Retrieval Augmented Generation. Jiang et al., EMNLP 2023.

[2] Synchronous Faithfulness Monitoring for Trustworthy Retrieval-Augmented Generation. Wu et al., EMNLP 2024.

[3] Lookback Lens: Detecting and Mitigating Contextual Hallucinations in Large Language Models Using Only Attention Maps. Chuang et al., EMNLP 2024.

**Questions:**

(please also see weaknesses)

* What is this paper’s definition of honesty? Can you further explain how it connects with adaptive retrieval in the proposed framework?

* I would like to see a full ablation on the proposed techniques, where only one technique is ablated at a time and compared to the performance of all the techniques together.

* How do you determine the hyperparameters of the proposed method and the baselines (such as the retrieval threshold)? Do you always use an in-domain validation dataset to tune the parameters? Are the hyperparameters kept the same across all test sets?

---

> ### Author Response · Authors · 2024-11-21
> **(1-1/7) Response to nBtC W1**
>
> We would like to thank the reviewer for their time and effort in reviewing our paper. We very much appreciate the insightful suggestions. We hereby address the concerns below:
>
> Reviewer ```nBtC``` **W1**: Most parts of the paper are (1) disconnected and (2) irrelevant to the key challenge of adaptive retrieval.
>
> **Response to W1**: We appreciate the reviewer's concern regarding the alignment between our proposed components and ARAG's core challenges. We would like to clarify several key points as follows:
>
> **Unified Framework with Interconnected Components**:
>
> We highlight that the objective of our entire solution design is to construct an ARAG method, where honesty steering, confidence monitoring, and search query formulation are essential, interconnected components. As stated in the Introduction (Lines 35–37), ARAG involves determining what and when to retrieve, with the design of the what aspect typically dependent on the construction of the when aspect [1][2][3]. Among these, honesty steering and confidence monitoring address the problem of when to retrieve, while search query formulation focuses on solving what to retrieve. Furthermore, same as all existing ARAG works [1][2][3], we evaluate retrieval timing effectiveness through the quality of final generated responses, as this is the most direct measure of an ARAG system's performance.
>
> **The Role of Honesty Steering**:
>
> Honesty steering plays a dual role that is directly relevant to both response quality and retrieval timing:
> 1. Aligning LLM output with its internal knowledge to avoid situations where the model "lies." This alignment improves the quality of LLM responses and simplifies the detection of retrieval timing, as the model's output more closely reflects its actual internal knowledge.
> 2. Encouraging LLM to acknowledge its limitations when applicable.
>
> For the first point, as discussed in Lines 46–57, ARAG fundamentally seeks to detect the boundaries of an LLM's knowledge, i.e., whether its parametric knowledge can directly answer a question. Mainstream ARAG methods typically rely on analyzing LLM outputs, such as response consistency, token probabilities, or prompting the LLM for reflection, etc. However, these methods are less accurate when the LLM's output deviates from its internal knowledge. Studies have shown that LLM responses sometimes exhibit hallucinations or dishonesty[5][6]. Therefore, we propose to use honesty steering to align LLM outputs with its internal knowledge. Our analysis in Figure 8 demonstrates that honesty steering effectively controls the honesty of LLM outputs.
>
> For the second point, honesty steering ensures that the LLM admits when it lacks knowledge or relevant information, as shown in Figure 10. Through this mechanism, when the LLM does not possess specific knowledge about a question or only irrelevant contents are provided, it acknowledges its limitations or explicitly states the absence of relevant knowledge, rather than resorting to speculation (i.e., "lying") or displaying overconfidence.

---

> ### Author Response · Authors · 2024-11-21
> **(1-2/7) Response to nBtC W1**
>
> Reviewer ```nBtC``` **W1**: Most parts of the paper are (1) disconnected and (2) irrelevant to the key challenge of adaptive retrieval.
>
> **Response to W1**:
>
> **Confidence-Based Retrieval Trigger**:
>
> We respectfully disagree that our retrieval triggering mechanism is uncertainty-based. Our approach differs fundamentally from uncertainty-based methods:
>
> 1. From the concept perspective, uncertainty-based methods typically measure consistency across multiple LLM responses (either through sampling or token probability analysis). In contrast, our approach leverages LLM's internal representations to assess its conceptual understanding, requiring no multiple sampling.
>
> 2. From the method perspective, our confidence monitoring doesn't employ probability or entropy calculations. Instead, it uses dot product operations to measure the similarity between LLM's internal representations and directional vector representing confidence.
>
> **Search Query Formulation as Essential Component**:
>
> Search query formulation, on the other hand, primarily addresses the issue of what to retrieve, which is also a critical component of the ARAG process. The general workflow of ARAG can be summarized as a "generate–halt generation–retrieve–resume generation" loop. When the LLM detects retrieval timing during the generation process, it pauses generation and retrieves relevant information. At this stage, a natural question arises: what content should be used as the search query? Options include using the most recent sentence, the entire history of generated content, the just-generated segment, prompting the LLM to summarize a query, etc. It is important to note that every ARAG method, such as Self-RAG [4], DRAGIN [2], SeaKR [3], and Rowen [1], has its own framework-specific search query formulation approach. Accordingly, we have also designed a dedicated search query formulation strategy tailored to our method.
>
> References:
>
> [1] Ding et al., Retrieve Only When It Needs: Adaptive Retrieval Augmentation for Hallucination Mitigation in Large Language Models, ArXiv 2024.
>
> [2] Su et al., DRAGIN: Dynamic Retrieval Augmented Generation based on the Information Needs of Large Language Models, ACL 2024.
>
> [3] Yao et al., SeaKR: Self-aware Knowledge Retrieval for Adaptive Retrieval Augmented Generation, ArXiv 2024.
>
> [4] Asai et al., Self-RAG: Learning to retrieve, generate, and critique through self-reflection, ICLR 2024.
>
> [5] Azaria et al., The Internal State of an LLM Knows When It's Lying, EMNLP 2023.
>
> [6] Slobodkin et al., The Curious Case of Hallucinatory Unanswerablity: Finding Truths in the Hidden States of Over-Confident Large Language Models, EMNLP 2023.

---

> ### Author Response · Authors · 2024-11-21
> **(2/7) Response to nBtC W2**
>
> Reviewer ```nBtC``` **W2**: The proposed method fails to trigger when the model confidently says “I don’t know”.
>
> **Response to W2**: Thank you for raising this important point. Our method actually addresses this issue through a hierarchical approach. The cornerstone of our approach is the Confidence Monitoring mechanism, which serves as the primary solution for detecting semantic unconfidence. Additionally, we have designed a Refusal Handling Module (RHM) as a complementary safeguard to address explicit refusal cases where the model directly states "I don't know" or similar responses. Due to space constraints in the main paper, we placed the detailed description of RHM in Appendix B.3.1 (ref. Line 1003-1025 and 1046-1073). We acknowledge that this component helps ensure the completeness of our method, and we have provided clear instructions in the main text, guiding readers to the location of the refusal handling module (ref. Line 219-225).
>
> The division of responsibilities between the confidence monitoring and refusal handling module is as follows:
>
> 1. In addition to the internal confidence level when generating the response, the confidence monitoring can effectively detect lack of confidence at both the linguistic and behavioral levels, where it can detect unconfident markers in expressions (e.g., "probably", "possibly", "it appears") and recognize hesitant or uncertain response patterns (e.g., nervous, unsure tones). As demonstrated in Figure 8, this multi-level detection ensures we catch subtle signs of unconfidence even when the model appears superficially confident.
>
> 2. The refusal handling module functions as a safety net, specifically focusing on detecting explicit refusal language in generated responses, such as direct admissions of lack of knowledge or missing relevant information.
>
> Through this hierarchical approach where confidence monitoring handles the primary semantic unconfidence detection and refusal handling provides additional coverage for explicit refusals, our method can effectively detect and appropriately respond to cases where the model expresses unconfidence or inability to answer, regardless of how confidently these responses are stated.

---

> ### Author Response · Authors · 2024-11-21
> **(3/7) Response to nBtC W3**
>
> Reviewer ```nBtC``` **W3**: Algorithm 1 is incomplete. An outer loop that calls algorithm 1 is required.
>
> **Response to W3**: We appreciate the reviewer's careful observation. Indeed, Algorithm 1 currently shows only the inner loop of our method, while an outer loop that coordinates the overall generation process is also necessary. The outer loop simply involves repeatedly calling this inner loop until the termination condition is met. We acknowledge this oversight and add the complete algorithm structure in the revised version (ref. Line 245-262 and Line 1018-1112).

---

> ### Author Response · Authors · 2024-11-21
> **(4/7) Response to nBtC W4**
>
> Reviewer ```nBtC``` **W4**: More citations to the adaptive RAG literature are required in section 3.
>
> **Response to W4**: Thank you for your advice. We have properly cited these work in our revised version (ref. Line 212-213 and 230).

---

> ### Author Response · Authors · 2024-11-21
> **(5/7) Response to nBtC Q1**
>
> Reviewer ```nBtC``` **Q1**: What is the definition of honesty and explain how it connects with adaptive retrieval.
>
> **Response to Q1**: We define honesty as the consistency between an LLM's responses and its internal belief [7]. This definition is crucial for our adaptive retrieval framework for several reasons:
>
> 1. The fundamental goal of Adaptive RAG (ARAG) is to detect the boundaries of an LLM's parametric knowledge, i.e., determining whether the model can directly answer a question using its internal knowledge.
> 2. Current ARAG methods primarily rely on analyzing model outputs through various approaches, like checking response consistency across multiple generations, analyzing token probabilities, using self-reflection prompts to let the model evaluate its own responses, etc. However, these methods become less reliable when there's a misalignment between the LLM's outputs and its internal knowledge. Studies have shown that LLMs sometimes exhibit behaviors where their responses deviate from their actual internal knowledge [5][6].
> 3. This is where honesty steering becomes essential: it encourages the LLM's outputs and its internal representation to better align with its internal knowledge (parametric knowledge). By doing so, we can more accurately and more easily detect when retrieval is truly necessary, rather than being misled by potentially unreliable surface-level responses.
> 4. Our empirical results (Figure 8) demonstrate that honesty steering effectively controls this alignment, leading to more reliable retrieval decisions in our ARAG framework.
>
> Reference:
>
> [7] Evans et al., Truthful AI: Developing and governing AI that does not lie, Arxiv 2021.

---

> ### Author Response · Authors · 2024-11-21
> **(6/7) Response to nBtC Q2**
>
> Reviewer ```nBtC``` **Q2**: Full ablation on the proposed techniques.
>
> **Response to Q2**: Thank you to the reviewer for pointing out this issue. First, we would like to clarify that we have conducted ablation studies for each module. Second, since the various modules in our method are deeply integrated, some modules (such as confidence monitoring and search query formulation) cannot be entirely decoupled from the framework. Therefore, while ensuring the framework's integrity, we have conducted a thorough analysis of each component:
>
> 1. **Feature Extraction**: Figure 6 examines the impact of different training datasets and varying dataset sizes on confidence feature extraction, where we demonstrate that the feature extraction is insensitive to the dataset selection and only requires few data samples to achieve good performance.
>
> 2. **Honesty Steering**: we evaluate the effectiveness of honesty steering from two perspectives:
> 2.1. In non-RAG setting, we investigate the validity of honesty steering and the effect of steering strength on TruthfulQA in Figure 2.
> 2.2. In RAG setting, we also measure the the effect of steering strength on both ASQA and PopQA in Figure 3. Meanwhile, we compare the performance of honesty steering, without honesty steering, and honesty-inducing prompts among PopQA, ASQA, and 2wiki datasets to demonstrate the effectiveness of honesty steering. Figure 5 further analyzes the impact of steering at different layers on CtrlA's performance
>
> 3. **Confidence Monitoring**: Since confidence monitoring is a fundamental component of Adaptive RAG, completely removing it would reduce our method to traditional RAG (SR-RAG, FS/FL-RAG). Therefore, we evaluate it through (1) comparative analysis with other Adaptive RAG methods (Tables 1-4), (2) investigation of threshold effects (Figure 4), and (3) isolated analysis of confidence feature effectiveness (Table 6).
>
> 4. **Search Query Formulation**: We present a comprehensive study of different search query formulation strategies across various retrievers in Table 7.
>
> We acknowledge that removing components individually could offer additional insights, but some, like confidence monitoring, are fundamental to Adaptive RAG. Nonetheless, our ablation studies already provide a comprehensive understanding of each component’s impact on system performance.

---

> ### Author Response · Authors · 2024-11-21
> **(7/7) Response to nBtC Q3**
>
> Reviewer ```nBtC``` **Q3**: How to determine the hyperparameters of the proposed method and the baselines.
>
> **Response to Q3**: For our proposed method, we selected the hyperparameter $\lambda$ through simple case studies. At the same time, the normalized confidence score falls within the range $[-1, 1]$, so we simply set $\tau$ to 0.0. For the compared baselines, we directly follow the parameter settings from prior work.
>
> We highlight that our method demonstrates strong robustness across different tasks and backbones without requiring task-specific hyperparameter selection. Thus,
> 1. We did not use any in-domain validation dataset for hyperparameter selection experiments.
> 2. As detailed in Lines 293-295 of our paper, we consistently use the same hyperparameters ($\lambda=0.3$ for honesty steering and $\tau=0.0$ for confidence threshold) across all experiments and different tasks.
> 3. The effectiveness of the fixed hyperparameters (without any task-specific adjustments) is evidenced by our strong performance across diverse tasks, including short-form QA, long-form QA, multi-hop QA, and FreshQA. Meanwhile, the performance among different backbones also demonstrates the robustness of our method.
>
> We believe it is a significant advantage of our method, as it eliminates the need for costly task-specific tuning and makes the method more practical for real-world applications. The reason for this robustness lies in our feature extraction strategy, which captures fundamental aspects of LLM cognition (honesty and confidence) that generalize well across different tasks and domains. As demonstrated in Figure 6, our method maintains stable performance even with varying amounts of data used for feature extraction, further supporting its inherent stability.

---

> ### Comment · Reviewer_nBtC · 2024-11-24
> **Thank you**
>
> Thank you for the detailed response. I decided to keep my ratings.

---

> > ### Author Response · Authors · 2024-11-25
> >
> > Dear Reviewer,
> >
> > Thank you again for your detailed and thoughtful feedback. We've tried our best to address the concerns raised in your feedback. Could you please confirm if any specific concerns still need clarification? If most concerns have been adequately addressed, we kindly request that you consider updating the rating to reflect these improvements.
> >
> > Best regards

---

### Official Review · Reviewer_kZpG · 2024-10-30

**Soundness:** 3
**Presentation:** 3
**Contribution:** 2
**Rating:** 5
**Confidence:** 3

**Summary:**

This paper tackles an issue in Adaptive Retrieval-Augmented Generation by focusing on retrieval timing to reduce hallucinations, and using interpretable methods to analyze honesty and confidence. The method shows promising experimental results. However, the motivation behind some methods feels unclear, with Honesty Steering and Search Query Formulation seeming unrelated to the core goal and lacking novelty. Additionally, the experimental setup lacks clarity on segment definitions. The reliance on an outdated linear representation assumption raises concerns about applicability in modern LLMs.

**Strengths:**

This paper proposes to address Adaptive Retrieval-Augmented Generation by focusing on retrieval timing to reduce hallucinations. It takes an interpretable approach by analyzing internal mechanisms like honesty and confidence, providing clearer insights into model behavior. Experimental results show meaningful improvements, suggesting that the proposed method contributes effectively to retrieval-augmented generation tasks.

**Weaknesses:**

1.	The paper doesn’t explain why the methods are crucial for Adaptive Retrieval-Augmented Generation (ARAG). The section Confidence Monitoring as Retrieval Trigger seems directly relevant to ARAG, while the section Honesty Steering and Search Query Formulation seems to be unrelated to the main goal. The Honesty Steering part has already been introduced in similar work[1], which makes the technical contribution of the paper incremental. The methods aren’t presented in a clear, motivating way, making the paper feel disjointed.

2.	In the experiments, it’s unclear how the "segments" are defined, leaving readers without a solid understanding of the setup. Also, the ablation study misses the marks that specifically show if Confidence Monitoring is driving the improvements. Without clear evidence, it’s hard to conclude that the approach tackles the core ARAG issues.

3.	The paper builds on a linear representation assumption, referencing work from Templeton et al. (2024) which is based on early RNN-based findings[2]. Given the evolution of model architectures in the LLM era, this assumption seems to be oversimplified.


[1] Li, Kenneth, et al. "Inference-time intervention: Eliciting truthful answers from a language model." Advances in Neural Information Processing Systems 36 (2024).
[2] Linguistic regularities in continuous space word representations." Proceedings of the 2013 conference of the north american chapter of the association for computational linguistics: Human language technologies. 2013.

**Questions:**

See the above comments.

---

> ### Author Response · Authors · 2024-11-21
> **(1-1/3) Response to kZpG W1**
>
> We would like to thank the reviewer for their time and effort in reviewing our paper. We very much appreciate the insightful suggestions. We hereby address the concerns below:
>
> Reviewer ```kZpG``` **W1**: The paper doesn’t explain why the methods are crucial for Adaptive Retrieval-Augmented Generation. The honesty steering part has already been introduced in similar work[1].
>
> **Response to W1**: **Relationship between our work and ARAG**
>
> Firstly, we highlight that the objective of our entire solution design is to construct an ARAG method, where honesty steering, confidence monitoring, and search query formulation are essential components. As stated in the Introduction (Lines 35–37), ARAG involves determining **what and when to retrieve**, with the design of the what aspect typically dependent on the construction of the when aspect [2][3][4]. Among these, honesty steering and confidence monitoring address the problem of when to retrieve, while search query formulation focuses on solving what to retrieve.
>
> The role of honesty steering is twofold:
> 1. Aligning LLM output with its internal knowledge to avoid situations where the model "lies." This alignment improves the quality of LLM responses and simplifies the detection of retrieval timing, as the model's output more closely reflects its actual internal knowledge.
> 2. Encouraging LLM to acknowledge its limitations when applicable.
>
> For the first point, as discussed in Lines 46–57, ARAG fundamentally seeks to detect the boundaries of an LLM's knowledge, i.e., whether its parametric knowledge can directly answer a question. Mainstream ARAG methods typically rely on analyzing LLM outputs, such as response consistency, token probabilities, or prompting the LLM for reflection, etc. However, these methods are less accurate when the LLM's output deviates from its internal knowledge. Studies have shown that LLM responses sometimes exhibit hallucinations or dishonesty [5][6]. Therefore, we propose to use honesty steering to align LLM outputs with its internal knowledge. Our analysis in Figure 7 demonstrates that honesty steering effectively controls the honesty of LLM outputs.
>
> For the second point, honesty steering ensures that the LLM admits when it lacks knowledge or relevant information, as shown in Figure 10. Through this mechanism, when the LLM does not possess specific knowledge about a question or only irrelevant contents are provided, it acknowledges its limitations or explicitly states the absence of relevant knowledge, rather than resorting to speculation (i.e., "lying") or displaying overconfidence.
>
> Search query formulation, on the other hand, primarily addresses the issue of what to retrieve, which is also a critical component of the ARAG process. The general workflow of ARAG can be summarized as a "generate–halt generation–retrieve–resume generation" loop. When the LLM detects retrieval timing during the generation process, it pauses generation and retrieves relevant information. At this stage, a natural question arises: what content should be used as the search query? Options include using the most recent sentence, the entire history of generated content, the just-generated segment, prompting the LLM to summarize a query, etc. It is important to note that every ARAG method, such as Self-RAG [7], DRAGIN [3], SeaKR [4], and Rowen [2], has its own framework-specific search query formulation approach. Accordingly, we have also designed a dedicated search query formulation strategy tailored to our method.

---

> ### Author Response · Authors · 2024-11-21
> **(1-2/3) Response to kZpG W1**
>
> Reviewer ```kZpG``` **W1**: The paper doesn’t explain why the methods are crucial for Adaptive Retrieval-Augmented Generation. The honesty steering part has already been introduced in similar work[1].
>
> **Response to W1**:
>
> **The Novelty of Honesty Steering**: we discuss the differences between our method (CtrlA) and ITI [1] from the following three perspectives:
>
> 1. From the task perspective: ITI and CtrlA aim to solve different problems. ITI focuses on enhancing the truthfulness of model responses through interventions, while CtrlA seeks to explore the directional features within the internal representations of LLMs and use these features to constrain and control the model's behavior to achieve dynamic retrieval. Moreover, CtrlA aims at bridging representational understanding and practical applications and has validated both the interpretability of the extracted directional vectors and their utility in complex RAG settings.
>
> 2. From the concept perspective: ITI emphasizes truthfulness, which refers to the alignment between an LLM's response and the ground truth. CtrlA, on the other hand, focuses on honesty, which is about the alignment between the LLM's response and its internal knowledge. In CtrlA, the goal is not necessarily to align the LLM's steered responses with the ground truth (as this alignment can be achieved through RAG). Instead, CtrlA aims to make the LLM's responses better reflect its internal cognition, enabling the detection of more accurate retrieval timing.
>
> 3. From the method perspective: ITI explores minimally invasive methods to intervene in model outputs to enhance "truthfulness." Specifically, it uses the TruthfulQA dataset to record the output activation vectors of the final token at each attention head across all layers. For a model with L layers and H heads per layer, this generates $L \times H$ activation vectors. These vectors are paired with corresponding labels to form supervised training and validation sets. ITI trains a detector (a single-layer fully connected network) for each attention head, then selects the top-K attention heads with the highest validation accuracy as intervention targets. During inference, an additional value is added to the output of these attention heads to control the LLM's output.
>
> In contrast, CtrlA is entirely unsupervised. It extracts directional features for specific concepts using contrastive prompts and PCA. Instead of intervening in attention heads, it targets the outputs of each LLM layer, ultimately extracting just one feature per layer. Additionally, CtrlA does not require classifiers for training. The method encompasses both steering and monitoring. Steering constrains the LLM’s output behavior to align it with its internal cognition while monitoring assesses the model’s output confidence to determine whether retrieval is necessary.
>
> Overall, ITI and CtrlA differ in both tasks and implementation methods. ITI's performance heavily depends on the quality of the training dataset, while CtrlA, based on instruction templates for feature extraction, is independent of the training dataset (see Figure 6). Furthermore, CtrlA demonstrates strong generalization across four backbone models and seven benchmark datasets since the extracted features are model-based rather than task-based.
>
> References:
>
> [1] Li et al., Inference-Time Intervention: Eliciting Truthful Answers from a Language Model, NeurIPS 2023.
>
> [2] Ding et al., Retrieve Only When It Needs: Adaptive Retrieval Augmentation for Hallucination Mitigation in Large Language Models, ArXiv 2024.
>
> [3] Su et al., DRAGIN: Dynamic Retrieval Augmented Generation based on the Information Needs of Large Language Models, ACL 2024.
>
> [4] Yao et al., SeaKR: Self-aware Knowledge Retrieval for Adaptive Retrieval Augmented Generation, ArXiv 2024.
>
> [5] Azaria et al., The Internal State of an LLM Knows When It's Lying, EMNLP 2023.
>
> [6] Slobodkin et al., The Curious Case of Hallucinatory Unanswerablity: Finding Truths in the Hidden States of Over-Confident Large Language Models, EMNLP 2023.
>
> [7] Asai et al., Self-RAG: Learning to retrieve, generate, and critique through self-reflection, ICLR 2024.

---

> ### Author Response · Authors · 2024-11-21
> **(2/3) Response to kZpG W2**
>
> Reviewer ```kZpG``` **W2**: Unclear how the "segments" are defined. The ablation study misses the marks that specifically show if Confidence Monitoring is driving the improvements.
>
> **Response to W2**: Sorry for the confusing description. The definition of “segments” refers to sentences, which are the same as in other ARAG methods and terminologies such as FLARE, Self-RAG, and DARGIN. In the revised version, we have used a more precise decsription to clarify this in the revised version (ref. Line 141-142).
>
> We highlight that confidence monitoring is the core part of our method because it is used to determine the retrieval timing. If confidence monitoring is directly removed, our method degrades into a traditional RAG approach. For a comparison between traditional RAG methods and our approach, please refer to the results of SR/FS/FL-RAG and CtrlA in Tables 1-4. As for the ablation studies on confidence monitoring, these are presented in Figure 4 and Table 6. Specifically, Figure 4 shows the performance changes of CtrlA under different thresholds, while Table 6 includes a set of answerable and unanswerable questions to test the accuracy of confidence monitoring detection.

---

> ### Author Response · Authors · 2024-11-21
> **(3/3) Response to kZpG W3**
>
> Reviewer ```kZpG``` **W3**: The paper builds on a linear representation assumption, referencing work from Templeton et al. (2024) which is based on early RNN-based findings[2].
>
> **Response to W3**: We agree with the reviewer’s opinion that Templeton et al. (2024) [8] builds upon earlier RNN-based findings [9]. However, recent works [10][11][12], including [8], are all based on transformer architectures from the LLM era and have reached similar conclusions. Our work is also based on these findings and we believe this approach is both reasonable and reliable. Furthermore, the Claude team in [8] has already validated this assumption in Claude 3 Sonnet, where they successfully extracted a large number of features representing specific concepts.
>
> References:
>
> [8] Templeton et al., Scaling Monosemanticity: Extracting Interpretable Features from Claude 3 Sonnet, 2024.
>
> [9] Mikolov et al., Linguistic Regularities in Continuous Space Word Representations, NAACL 2013.
>
> [10] Slobodkin et al., The Curious Case of Hallucinatory Unanswerablity: Finding Truths in the Hidden States of Over-Confident Large Language Models, EMNLP 2023.
>
> [11] Marks et al., The Geometry of Truth: Emergent Linear Structure in Large Language Model Representations of True/False Datasets, ArXiv 2023.
>
> [12] Liu et al., Aligning Large Language Models with Human Preferences through Representation Engineering, ACL 2024.

---

> ### Author Response · Authors · 2024-11-25
>
> Dear reviewer,
>
> Thank you again for your detailed feedback. We have tried to address your concerns by
>
> - further explaining the relationship between our work and ARAG
> - referencing existing sections and experimental results in our manuscript
> - citing relavent literature .
>
> We would appreciate your thoughts on whether these clarifications address your concerns, or if any points need further elaboration.
>
> Thank you for your time and consideration.

---

> ### Comment · Reviewer_kZpG · 2024-11-26
> **Thank you for the detailed responses.**
>
> I acknowledge that I have read the responses and would like to keep the scores.

---

> > ### Author Response · Authors · 2024-11-26
> >
> > Dear Reviewer,
> >
> > We greatly appreciate your comprehensive review and insightful comments. We have tried our best to address each point you raised. Have we addressed your main concerns? If so, we would appreciate consideration for a raised rating based on the improvements made. If not, we would be grateful if you could let us know which concerns remain unaddressed.
> >
> > Thank you for your time and consideration.

---

### Official Review · Reviewer_s1Gu · 2024-10-31

**Soundness:** 3
**Presentation:** 3
**Contribution:** 3
**Rating:** 5
**Confidence:** 4

**Summary:**

This paper proposes CtrlA, an adaptive RAG framework to extract honesty and confidence features and use them to steer the generation. CtrlA uses contrastive prompts to instruct the LLM with corresponding behaviors (honest/dishonest/confident/unconfident), and then extracts the hidden representations of the whole statement to form the honesty/confidence vectors with PCA. It uses a linear combination to guide the generation towards the honest/confident direction. Experimental results and analyses demonstrate that CtrlA performs well in reasoning tasks.

**Strengths:**

- The proposed method is interesting and effective.

- The paper is well-written and easy to follow.

**Weaknesses:**

- The method requires accessing the internal structure of LLMs, which may not be available for some competitive commercial LLMs such as o1 and limits its applicability.

- The approach relies heavily on PCA and detailed layer-wise feature extraction, potentially leading to high computational costs and processing time, especially with larger models. The principle component selection process is also not discussed since the first principle component might not represent the required direction.

- The method involves intricate processes such as feature extraction, honesty steering, and confidence monitoring, which require substantial computational resources and expertise to implement. No token expenses are reported.

- The query formulation strategies lack clarity in handling ambiguous or noisy outputs. The methods for masking unconfident tokens and refining search queries are insufficiently detailed, making it challenging to understand how they ensure the retrieval of relevant documents. Additionally, there's a potential risk of losing important context when masking tokens, which could affect retrieval accuracy.

- Modifying the representation space can inherently harm the quality of generated content, which successfully answers the question but fails to produce a plausible and fluent sentence, as illustrated in Fig 3b.

- The experimental results are not comprehensively discussed. Fig 2 and 3 show different effects of $\lambda$ on accuracy, but no further discussion is provided for such contradiction except the dataset category.

**Questions:**

- Why does honesty steering always improve accuracy in closed-domain QA? Generally, the result should be similar to open-domain tasks.

---

> ### Author Response · Authors · 2024-11-21
> **(1/6) Response to s1Gu W1**
>
> We would like to thank the reviewer for their time and effort in reviewing our paper. We very much appreciate the insightful suggestions. We hereby address the concerns below:
>
> Reviewer ```s1Gu``` **W1**: The method may not be available for some competitive commercial LLMs such as o1 and limits its applicability.
>
> **Response to W1**: First, we agree with the reviewer's comments that our method cannot be directly applied to proprietary LLMs, as we do not have access to their internal structures. However, as our primary goal is to explore Adaptive RAG (ARAG) from a different perspective and try to solve this problem. While methods based on model output probabilities or verbalized feedback can indeed be applied to both proprietary and open-sourced LLMs, these are not the focus of our research. Our approach aims to investigate how internal representations of the model reflect specific states of the LLM and, based on this, to develop our ARAG strategies. We believe this technical route can also offer valuable insights for proprietary LLMs and inspire potential improvements.

---

> ### Author Response · Authors · 2024-11-21
> **(2/6) Response to s1Gu W2**
>
> Reviewer ```s1Gu``` **W2**: The approach relies on PCA and layer-wise feature extraction, potentially leading to high computational costs and processing time, especially with larger models.
>
> **Response to W2**: **Complexity of Honesty and Confidence Feature Extraction**
>
> We highlight that our method does not require recalculating honesty and confidence features for every generation. These features are extracted and computed offline using a small amount of data (as detailed in Lines 175–177 and Appendix C.1, Lines 1118–1150). During the entire inference process and across experiments on different datasets, the honesty and confidence features are directly used without any additional operations. The effectiveness and transferability of these extracted features are discussed in detail in Section 5.2 (Lines 371–415).
>
> Additionally, as shown in Figure 6, we demonstrate that our feature extraction method is insensitive to the choice of datasets and does not require large datasets. Features extracted from as few as 512 tokens already achieve excellent results. Encoding these 512 tokens (approximately 50 sentences) using LLM and applying PCA to extract features is very fast.
>
> Our experimental results on Mistral-7B show that the total time for feature collection and PCA-based extraction on a single NVIDIA Tesla V100 32GB GPU is approximately **1.12 minutes**. This demonstrates that our method is highly efficient.
>
> We agree with the reviewer that larger models will incur higher computational costs and longer processing times. However, the overhead of PCA and layer-wise feature extraction is acceptable and is almost negligible compared to the overall cost of LLM inference on RAG datasets.
>
> **Principle Component Selection Process**
> Strictly speaking, we cannot directly measure whether a specific direction corresponds to a particular concept. Existing works[1][2][3][4] are not able to directly determine this either, as we lack both a ground-truth anchor (i.e., a ground-truth directional vector representing a specific concept) and a unified evaluation metric. However, we can infer the reliability of the extracted features from experimental results:
> 1. The TruthfulQA dataset is designed to evaluate the truthfulness of LLM responses or the extent to which LLMs can honestly answer questions. On TruthfulQA (Figure 2), by increasing the strength of honesty steering within a certain range, the model can be prompted to provide more truthful answers. This demonstrates the validity of the extracted honesty feature.
> 2. To verify the reliability of the confidence feature, we constructed a dataset of answerable and unanswerable questions (see construction method in Appendix C.1, Lines 1130–1150) to evaluate whether the extracted confidence feature can accurately reflect the model’s confidence when answering different questions. As shown in Table 6, confidence monitoring accurately detected the model’s internal confidence levels during responses. Answerable questions exhibited high confidence, while unanswerable questions showed low confidence.
> 3. We conducted a series of case studies to verify the effectiveness of the extracted honesty and confidence features. Specifically, we used these features to influence the behavior of the LLM, prompting it to generate outputs aligned with a specific concept and observing whether the LLM’s responses followed the desired direction. As shown in Figures 7, 8, 9, and 10 in the Appendix, the extracted honesty and confidence features effectively intervened in and controlled the generation behavior of the LLM, demonstrating the relevance of these directional features to Honesty and Confidence.
> 4. Additionally, we performed detailed ablation experiments on the Honesty and Confidence features, further proving their effectiveness (see Figure 3, Figure 4, and Table 5).
>
> References:
>
> [1] Elhage et al., Toy Models of Superposition, ArXiv 2022.
>
> [2] Slobodkin et al., The Curious Case of Hallucinatory Unanswerablity: Finding Truths in the Hidden States of Over-Confident Large Language Models, EMNLP 2023.
>
> [3] Marks et al., The Geometry of Truth: Emergent Linear Structure in Large Language Model Representations of True/False Datasets, ArXiv 2023.
>
> [4] Templeton et al., Scaling Monosemanticity: Extracting Interpretable Features from Claude 3 Sonnet, 2024.

---

> ### Author Response · Authors · 2024-11-21
> **(3/6) Response to s1Gu W3**
>
> Reviewer ```s1Gu``` **W3**: The method requires substantial computational resources and expertise to implement.
>
> **Response to W3**: It is important to clarify that the honesty and confidence features we extract are model-based rather than task-based. Once these features are extracted initially, they remain unchanged. Across all tasks, datasets, and experiments, we consistently use the same features. Additionally, as W2 mentioned, our feature extraction method is very simple and requires a little time.
>
> For the complexity of honesty steering and confidence monitoring during inference, our method does not significantly increase computational costs during inference. For LLMs, the computational complexity of each Transformer layer is approximately $O(n^2d + nd^2)$. In contrast, Honesty Steering and Confidence Monitoring only involve simple vector addition, subtraction, and dot product operations on the representations output by each layer of the LLM, with a computational complexity of just $O(nd)$.
>
> Therefore, both in training and inference, the additional overhead introduced by our method is relatively minimal. On the training side, methods like Self-RAG [5] and QC-RAG [6] require large amounts of curated SFT data to fine-tune the LLM or train an additional classifier. In contrast, our feature extraction takes only about one minute. On the inference side, methods like Rowen [7] and the concurrent work SeaKR [8] require generating dozens of answers per query to evaluate their consistency, whereas we only need a single path generation.
>
> Regarding token expenses, we emphasize that calculating token expenses is meaningless for our method. Honesty steering and confidence monitoring do not generate new tokens but instead operate on the representations output for each token by the LLM.
>
> References:
>
> [5] Asai et al., Self-RAG: Learning to retrieve, generate, and critique through self-reflection, ICLR 2024.
>
> [6] Jeong et al., Adaptive-RAG: Learning to adapt retrieval-augmented large language models through question complexity, NAACL 2024.
>
> [7] Ding et al., Retrieve Only When It Needs: Adaptive Retrieval Augmentation for Hallucination Mitigation in Large Language Models, ArXiv 2024.
>
> [8] Yao et al., SeaKR: Self-aware Knowledge Retrieval for Adaptive Retrieval Augmented Generation, ArXiv 2024.

---

> ### Author Response · Authors · 2024-11-21
> **(4/6) Response to s1Gu W4**
>
> Reviewer ```s1Gu``` **W4**: The query formulation strategies lack clarity in handling ambiguous or noisy outputs.
>
> **Response to W4**: The details of the query formulation are included in the appendix due to space constraints (Ref. Line 240 and Appendix B.2 Lines 959-999). To avoid the potential risk of losing important context when masking tokens, we retain old information and only mask new information (as LLMs generate content from left to right, old information is assumed to have already been verified for correctness). Additionally, we experimented with using a prompt-based LLM to generate queries (Ref. Lines 264-269 and Lines 969-999). Detailed ablation results for the query formulation strategy are reported in Table 7.
>
> In Section 3.2.4 (Lines 228-269) and Appendix B.2 (Lines 969-999), we elaborate on the design of the query formulation. Specifically, we propose to use the “new information” of the generated segment $y_t$ as the search query for retrieval. The "new information" denotes the tokens that do not appear in both input $x$ and preceding generations $\hat{y}<t$. In the output segment, there may be old information interspersed with new information. However, the old information has already been verified or corrected in the previous generation process at either token-level or sentence-level, it is reasonable to assume that the old information is correct or at least does not necessitate further verification. Besides, the confidence monitoring is not always accurate in pinpointing specific tokens and may identify "unconfident" tokens at trivial positions, such as stopwords. Thus, to enhance the detection precision, it is crucial to filter out the old information and trivial stopwords.

---

> ### Author Response · Authors · 2024-11-21
> **(5/6) Response to s1Gu W5**
>
> Reviewer ```s1Gu``` **W5**: Modifying the representation space can inherently harm the quality of generated content.
>
> **Response to W5**: We believe this result is a normal manifestation because it is impossible to infinitely increase the perturbation to an LLM. Overall, our method improves the LLM's specific capabilities and performance by modifying its representation space within a certain range. However, excessive perturbation of the LLM's representation inevitably disrupts its semantic space, leading to a performance decline.
>
> For instance, suppose the LLM's representation space is $R = (0, 0, 1)$ and our feature is $v = (1, 0, 0)$. After steering, the LLM’s representation becomes $R+\alpha v=(\alpha, 0, 1)$. When $\alpha$ is too large, such as $\alpha=1000$, the representation becomes $R+\alpha v=(1000,0,1)$, where the LLM's representation vector is completely dominated by $\alpha$ and $v$.
>
> Additionally, excessive modification of the LLM’s representation space can lead to performance degradation. This issue is also encountered in fields like knowledge editing [9][10] or other studies [11][12]. However, for our method, we avoid this issue by controlling the hyperparameters:
> 1. In all major experiments, we consistently used a relatively small coefficient of $\lambda = 0.3$, which achieved significant performance improvements.
> 2. The MAUVE metric in ASQA, to some extent, reflects the fluency of the generated content (ref. Table 2 and Table 5). It can be observed that using honesty steering simultaneously improves the MAUVE score.
>
> References:
>
> [9] Wang et al., EasyEdit: An Easy-to-use Knowledge Editing Framework for Large Language Models, ACL 2024.
>
> [10] Gu et al., Model Editing Harms General Abilities of Large Language Models: Regularization to the Rescue, EMNLP 2024.
>
> [11] Li et al., Inference-Time Intervention: Eliciting Truthful Answers from a Language Model, NeurIPS 2023.
>
> [12] Yin et al., LoFiT: Localized Fine-tuning on LLM Representations, ArXiv 2024.

---

> ### Author Response · Authors · 2024-11-21
> **(6/6) Response to s1Gu W6&Q1**
>
> Reviewer ```s1Gu``` **W6**: Fig 2 and 3 show different effects of $\lambda$ on accuracy.
>
> Reviewer ```s1Gu``` **Q1**: Why does honesty steering always improve accuracy in closed-domain QA?
>
> **Response to W6**: Regarding the inconsistencies caused by $\lambda$ in Figures 2 and 3, as well as the observation that honesty steering always improves performance in closed-domain QA (TruthfulQA), this is due to our lack of further testing with more $\lambda$ values. In reality, the performance and trend of honesty steering in closed-domain QA are consistent with those in open-domain QA, showing an initial rise followed by a decline, with the difference being in the range of $\lambda$. This phenomenon has already been explained in our response in W5, where we noted that it is impractical to infinitely increase perturbation.
>
> To validate this conclusion, we tested additional, larger values of $\lambda$ on TruthfulQA, with the results as follows (we also refine the figure in the revised version (ref. Line 378-387) and the observations (ref. Line 374-377, 402-404, 429-431)):
>
> | $\lambda$ | 0.0  | 0.3  | 0.5  | 0.7  | 1.0  | 1.5  | 2.0  | 2.5  |
> |-----------|-------|-------|-------|-------|-------|-------|-------|-------|
> | Accuracy  | 41.30 | 41.44 | 42.16 | 44.00 | 46.20 | 49.26 | 45.90 | 40.39 |
>
> The difference in $\lambda$'s range arises mainly due to the nature of the two types of tasks. For TruthfulQA, the LLM only needs to output a small number of tokens in the final response without retrieved contents, such as “A”, “B”, “C” and “D”. Thus, in this task, the responses are shorter, and the vocabulary space is much smaller (e.g., only choosing from the given options). For such a simple task, the LLM has a higher tolerance for $\lambda$ values, and positive gains can be achieved within the acceptable range of $\lambda$.
>
> In contrast, for open-domain QA tasks, the LLM incorporates a large amount of external information through retrieval, and its responses are longer (usually a sentence or multiple sentences). Additionally, the search space for decoding each token encompasses the entire vocabulary (since the exact words to output cannot be predetermined). Under such circumstances, the LLM’s tolerance for $\lambda$ is lower, and the acceptable range is narrower.

---

> > ### Comment · Reviewer_s1Gu · 2024-11-25
> >
> > Thank you for the detailed responses. I acknowledge that I have read the responses and would like to keep the rating.

---

> > > ### Author Response · Authors · 2024-11-25
> > >
> > > Dear Reviewer,
> > >
> > > Thank you again for your thorough feedback. We've tried our best to address the concerns you raised and would appreciate knowing if any concerns remain. If our improvements meet your expectations, we'd be grateful if you could update the rating accordingly.
> > >
> > > Best regards

---

### Official Review · Reviewer_FtsS · 2024-11-01

**Soundness:** 2
**Presentation:** 4
**Contribution:** 2
**Rating:** 3
**Confidence:** 4

**Summary:**

This paper proposes an adaptive RAG framework, CTRLA, which focuses on dynamically activating retrieval to supplement LLMs only when needed. Instead of relying solely on statistical confidence measurements, the paper introduces a representation-based approach to extract "honesty" and "confidence" features that guide the retrieval process. This framework targets issues of hallucination and efficiency in RAG models by enabling more accurate timing of retrievals. The experiments demonstrate that CTRLA outperforms other adaptive RAG methods, showcasing its effectiveness in retrieval timing and content accuracy.

**Strengths:**

1. CTRLA introduces a method for adaptive retrieval, focusing on honesty and confidence features within the LLM's representation space, a departure from conventional uncertainty-based approaches.
2. The experiments reveal CTRLA’s effectiveness in generating accurate and relevant responses, with enhanced performance in both short-form and long-form QA tasks.
3. The approach optimizes retrieval timing, resulting in fewer unnecessary retrievals.
4. The CTRLA framework is lightweight and does not heavily depend on external fine-tuning, potentially making it applicable across various LLMs and use cases.
5. The figures in the paper are very exquisite.

**Weaknesses:**

1. The experiments cover several QA tasks, and additional datasets that may not be memorized by the LLM could have further validated the framework's robustness, especially across rapidly changing topic datasets.
2. There are many similar works described as "knowledge conflicts", but the author seems to miss them both for introducing and as baselines, such as 2.1 Xie, Jian, et al. "Adaptive chameleon or stubborn sloth: Revealing the behavior of large language models in knowledge conflicts." arXiv preprint arXiv:2305.13300 (2023). 2.2 Jin, Zhuoran, et al. "Tug-of-war between knowledge: Exploring and resolving knowledge conflicts in retrieval-augmented language models." arXiv preprint arXiv:2402.14409 (2024).
3. Does this multi-step pipeline approach increase the computational burden? What is the overall complexity?
4. Since CTRLA relies on inherent LLM honesty and confidence features, the effectiveness of the framework may vary significantly depending on the LLM's underlying architecture.
5. This paper is based on a strong assumption that "honesty" and "confidence" features can be linearly separated within the LLM's latent space. Does the independence between the two still hold in complex contexts?
6. The extraction of "honesty" and "confidence" is based on the parameterized knowledge of the model itself. How to ensure the fairness of this process? Because this is the basis for the following work.

**Questions:**

See above

---

> ### Author Response · Authors · 2024-11-21
> **(1/6) Response to Reviewer FtsS W1**
>
> We would like to thank the reviewer for their time and effort in reviewing our paper. We very much appreciate the insightful suggestions. We hereby address the concerns below:
>
> Reviewer ```FtsS``` **W1**: Additional datasets that may not be memorized by the LLM could have further validated the framework's robustness.
>
> **Response to W1**: As mentioned in the Experiment Setup section (Lines 284–291), we conducted experiments on a total of seven datasets: Short-form QA (TriviaQA and PopQA), Long-form QA (ASQA and Bio), MultiHop QA (2WikiMultihopQA and HotpotQA), and FreshQA [1]. Among these, Short-form, Long-form, and MultiHop QA are standard benchmark datasets widely used by many RAG methods, while FreshQA [1] is a high-quality, real-time updated dataset designed to provide a reliable benchmark for LLMs.
>
> FreshQA includes questions related to dynamically changing knowledge points, such as recent events, technological developments, economic trends, and more. It is designed to evaluate models’ ability to handle tasks requiring up-to-date factual knowledge, dynamic reasoning, and contextual inference. Specifically, in our experiments, we used the version of FreshQA released on **April 8, 2024 (Lines 290–291)**, and all backbones employed in our experiments were released before this date. Under this setting, LLM can not solve questions in FreshQA via memorization.
>
> The evaluation results on FreshQA are shown in Table 4, where our method outperforms the compared baselines, demonstrating its robustness on datasets involving rapidly changing topics.
>
> Reference:
>
> [1] Vu et al., FreshLLMs: Refreshing Large Language Models with Search Engine Augmentation, ACL 2024.

---

> ### Author Response · Authors · 2024-11-21
> **(2/6) Response to FtsS W2**
>
> Reviewer ```FtsS``` **W2**: There are many similar works described as "knowledge conflicts", but the author seems to miss them both for introducing and as baselines.
>
> **Response to W2**: We would like to clarify that "Knowledge Conflicts" and "Adaptive RAG (ARAG) are **related but distinct** topics. In this paper, we mainly focus on the core issue of when to retrieve and what to retrieve for ARAG.
>
> 1. Knowledge Conflicts: This line of work primarily focuses on **analyzing the behavior of LLMs in scenarios where there are conflicts between external knowledge contents and the LLM's internal knowledge** by **giving the external contents**. Key aspects include how the model selects between internal and external knowledge, its robustness, and methods to mitigate knowledge conflicts.
> 2. Adaptive RAG (ARAG): This line of work mainly focuses on **determining, for a given query: (1) whether the LLM needs to retrieve external information (i.e., whether the model's internal knowledge is sufficient to answer the question) and (2) dynamically deciding during generation whether additional retrieval is necessary to assist in answering the question**.
>
> To clearly differentiate between these two types of work, we have cited the relevant papers on knowledge conflicts and provide a detailed discussion in the revised version (ref. Line 934-956).
>
> Regarding incorporating knowledge conflicts into baselines, we highlight that our work primarily focuses on the exploration of **when to retrieve and what to retrieve**. In contrast, research on knowledge conflicts primarily centers on analyzing the phenomenon and the behavior of LLMs when facing knowledge conflicts between external contents and LLM's internal knowledge, rather than directly focusing on RAG. Moreover, studies such as [3] often utilize specifically curated datasets and pre-specified external knowledge contents to simulate how models utilize knowledge at the "post-retrieval" stage, which diverges from the focus of our work, making direct cross-study performance comparisons challenging.
>
> We believe that Knowledge Conflicts and ARAG are complementary. We leave incorporating the concept of knowledge conflicts to enhance ARAG for future work. For instance, for queries identified by ARAG as requiring retrieval, we can further analyze whether the retrieved content conflicts with the model's internal knowledge and attempt to resolve such conflicts.
>
> References:
>
> [2] Xie et al., Adaptive Chameleon or Stubborn Sloth: Revealing the Behavior of Large Language Models in Knowledge Conflicts, ICLR 2024.
>
> [3] Jin et al., Tug-of-War Between Knowledge: Exploring and Resolving Knowledge Conflicts in Retrieval-Augmented Language Models, LREC 2024.

---

> ### Author Response · Authors · 2024-11-21
> **(3-1/6) Response to FtsS W3**
>
> Reviewer ```FtsS``` **W3**: Does this multi-step pipeline approach increase the computational burden? What is the overall complexity?
>
> **Response to W3**: It is important to note that RAG is a relatively complex system, and the components of various existing RAG methods are different, making it difficult to compare them directly. Therefore, we approach the comparison of our method with others by focusing on the method itself and the data, evaluating aspects such as complexity and computational burden.
>
> **Complexity of Retrieval and Generation**
>
> We would like to clarify that **all Adaptive RAG (ARAG) methods are based on a multi-step pipeline approach**.
> 1. Compared to ARAG methods requiring querying LLM multiple times, such as Rowen[4] and SeaKR[5], these methods require generating dozens of answers per query to evaluate their consistency and determine whether retrieval is necessary. In contrast, our method **only requires a single end-to-end inference**. During generation, the model dynamically decides whether retrieval is needed based on its internal confidence. This significantly reduces inference costs while achieving better results.
> 2. Compared to uncertainty-based ARAG methods, such as DRAGIN[6], DRAGIN computes three different scores for each generated token: entropy, attention values, and binary semantic indicator. Among these, calculating attention values involves quadratic complexity. In contrast, our method **only requires a single vector addition and multiplication operation per token**. Our method ensures higher efficiency and achieves better results compared to DRAGIN (see Tables 3 and 9). Furthermore, as reported in Table 3, our method achieves a lower average retrieval frequency on evaluation datasets compared to DRAGIN and FL/FS-RAG methods.
>
> From a dataset perspective:
> 1. For short-form QA (e.g., TriviaQA and PopQA), typically only one retrieval is needed. Unlike traditional RAG, which retrieves every question, our method dynamically determines during response generation whether retrieval is necessary. For questions where retrieval is unnecessary, the model can directly answer without retrieval. This reduces complexity since we only retrieve a subset of questions. The table below shows the average retrieval frequency on the TriviaQA dataset. SR-RAG retrieves every question (100% retrieval frequency), while CtrlA achieves a frequency of 84.8%, meaning 15.2% of questions are answered without retrieval. Although our retrieval frequency has decreased, the final results have significantly improved. We attribute this to two factors: (1) Honesty steering enables the model to more accurately output its internal knowledge; (2) Instead of performing retrieval before answering the question, we use confidence monitoring during the LLM’s response generation process to dynamically determine whether retrieval is needed. Combined with search query formulation, we can more accurately retrieve knowledge that the model doesn't know, thereby enhancing the final results.
>
>
> | **Method** | **Accuracy** | **Retrieval Frequency** |
> |------------|--------------|-------------------------|
> | SR-RAG     | 62.7         | 100%                   |
> | CtrlA      | 76.4         | 84.8%                  |
>
> 2. For long-form and multi-hop QA, which often require multiple retrievals to gather all necessary information, our method also dynamically determines retrieval timing during generation. As shown in Figure 1, when generating content about Parker's birthplace, our method detects the LLM's low confidence, reformulates the query, and retrieves specific information about Parker's birthplace. Overall, while all Adaptive RAG methods introduce additional complexity by incorporating retrieval and generation processes, CtrlA achieves equal or lower complexity compared to other ARAG methods while delivering superior performance. As shown in Table 3, our method outperforms DRAGIN[6] significantly while maintaining a similar or lower retrieval frequency. Additionally, the concurrent work SeaKR[5], an uncertainty-based approach, requires generating 20 responses for each question and calculating a Gram Matrix of these representations to decide whether retrieval is needed. In contrast, CtrlA only generates once.
>
> References:
>
> [4] Ding et al., Retrieve Only When It Needs: Adaptive Retrieval Augmentation for Hallucination Mitigation in Large Language Models, ArXiv 2024.
>
> [5] Yao et al., SeaKR: Self-aware Knowledge Retrieval for Adaptive Retrieval Augmented Generation, ArXiv 2024.
>
> [6] Su et al., DRAGIN: Dynamic Retrieval Augmented Generation based on the Information Needs of Large Language Models, ACL 2024.

---

> ### Author Response · Authors · 2024-11-21
> **(3-2/6) Response to FtsS W3**
>
> Reviewer ```FtsS``` **W3**: Does this multi-step pipeline approach increase the computational burden? What is the overall complexity?
>
> **Response to W3**: **Complexity of Honesty and Confidence Feature Extraction**
>
> Our method does not require recalculating honesty and confidence features for every generation. **These features are extracted and computed offline using a small amount of data (as detailed in Lines 175–177 and Appendix C.1, Lines 1118–1150)**. During the entire inference process and across experiments on different datasets, the honesty and confidence features are directly used without any additional operations. The effectiveness and transferability of these extracted features are discussed in detail in Section 5.2 (Lines 371–415).
>
> Additionally, as shown in Figure 6, we demonstrate that our feature extraction method is insensitive to the choice of datasets and does not require large datasets. Features extracted from as few as 512 tokens already achieve excellent results. Encoding these 512 tokens (approximately 50 sentences) using LLM and applying PCA to extract features is very fast.
>
> Our experimental results on Mistral-7B show that the total time for feature collection and PCA-based extraction on a single NVIDIA Tesla V100 32GB GPU is approximately **1.12 minutes**. This demonstrates that our method is highly efficient.
>
> **Complexity of Honesty Steering and Confidence Monitoring Inference**
>
> Our method does not significantly increase computational costs during inference. For LLMs, the computational complexity of each Transformer layer is approximately $O(n^2d + nd^2)$. In contrast, Honesty Steering and Confidence Monitoring only involve simple vector addition, subtraction, and dot product operations on the representations output by each layer of the LLM, with a computational complexity of just $O(nd)$.

---

> ### Author Response · Authors · 2024-11-21
> **(4/6) Response to FtsS W4**
>
> Reviewer ```FtsS``` **W4**: The effectiveness of the framework may vary significantly depending on the LLM's underlying architecture.
>
> **Response to W4**: Our method is minimally influenced by the LLM's architecture and is largely independent of the specific design of the LLMs. This is because our operational space is the representations output by each layer of the LLM, and we only need access to these representations. In addition to Mistral-7B, we conducted experiments on three different backbone LLMs (refer to Table 9). The results show that our method achieved significant improvements across all the backbone LLMs.

---

> ### Author Response · Authors · 2024-11-21
> **(5/6) Response to FtsS W5**
>
> Reviewer ```FtsS``` **W5**: This paper is based on a strong assumption that "honesty" and "confidence" features can be linearly separated within the LLM's latent space. Does the independence between the two still hold in complex contexts?
>
> **Response to W5**: We highlight that this assumption was proposed and validated by prior works[7][8][9][10], we are motivated by their observations. Many prior works have introduced and demonstrated its validity through extensive experiments [7][8][9][10], namely, that conceptual features can be effectively separated. Moreover, the Claude team has already validated this assumption in Claude 3 Sonnet, where they successfully extracted a large number of features representing specific meanings [10].
>
> References:
>
> [7] Elhage et al., Toy Models of Superposition, ArXiv 2022.
>
> [8] Slobodkin et al., The Curious Case of Hallucinatory Unanswerablity: Finding Truths in the Hidden States of Over-Confident Large Language Models, EMNLP 2023.
>
> [9] Marks et al., The Geometry of Truth: Emergent Linear Structure in Large Language Model Representations of True/False Datasets, ArXiv 2023.
>
> [10] Templeton et al., Scaling Monosemanticity: Extracting Interpretable Features from Claude 3 Sonnet, 2024.

---

> ### Author Response · Authors · 2024-11-21
> **(6/6) Response to FtsS W6**
>
> Reviewer ```FtsS``` **W6**: The extraction of "honesty" and "confidence" is based on the parameterized knowledge of the model itself. How to ensure the fairness of this process?.
>
> **Response to W6**: We validated the fairness of the extracted features from different dimensions:
> 1. On TruthfulQA (Figure 2), by increasing the strength of honesty steering within a certain range, the model can be prompted to output more truthful answers.
> 2. On our constructed answerable and unanswerable questions dataset (Table 6), confidence monitoring accurately detected the model's internal confidence when responding. Answerable questions exhibited high confidence, while unanswerable questions showed low confidence.
> 3. Additionally, we conducted a series of case studies to verify the effectiveness of the extracted honesty and confidence features. Specifically, we used the extracted features to attempt to alter the behavior of the LLM, inducing it to lean towards generating outputs aligned with a specific concept and observed whether the LLM produced outputs in the desired direction. Referring to the results in Figures 7, 8, 9, and 10 in the Appendix, the extracted honesty and confidence features were able to accurately intervene in and control the generation behavior of the LLM, demonstrating the effectiveness of these directional features.

---

> > ### Author Response · Authors · 2024-11-25
> >
> > Dear reviewer,
> >
> > Thank you again for your detailed feedback. We have tried to address your concerns by
> >
> > - citing relevant literature
> > - referencing existing sections in our manuscript
> > - providing new experimental results, including timing analysis for CtrlA feature extraction and complexity analysis for Honesty Steering and Confidence Monitoring.
> >
> > We would appreciate your thoughts on whether these clarifications address your concerns, or if any points need further elaboration.
> >
> > Thank you for your time and consideration.

---

> ### Comment · Reviewer_FtsS · 2024-11-25
>
> Thanks for the authors' feedback and I will keep my rate.

---

> > ### Author Response · Authors · 2024-11-26
> >
> > Dear Reviewer,
> >
> > Thank you for your thorough review. We've tried our best to address your concerns and would welcome your thoughts on whether the revisions and clarifications have resolved your concerns. We'd welcome an updated rating if the changes meet your expectations or additional feedback on outstanding concerns.
> >
> > Thank you for your time and consideration.

---

### Author Response · Authors · 2024-11-21
**General Response**

Dear Reviewers:

Thank you for your thorough and constructive feedback on our manuscript. We sincerely appreciate your time and effort in reviewing our work. We have carefully addressed all the comments and suggestions, which have significantly improved the quality of our paper.

Below is the summary of the major revisions we have made during the rebuttal:

1. Extended Experimental Analysis
   - Conducted and analyzed additional experiments on Honesty Steering with different coefficients using the TruthfulQA dataset (Section 5.2)
   - Enhanced our findings with detailed analysis (lines 374-375, 385-386, 402-404, 429-431)
   - Revised visual representation of results (Figure 2)
2. Enhanced Theoretical Framework and Algorithm Description
   - Added explanations and references about output segment processing (Section 3.1, line 137,  line 142)
   - Improved presentation of Algorithm 1 (Section 3.2.4 lines 245-262) and Algorithm 3 (Appendix B 3.2 lines 1089-1112 )
   - Clarified the role of confidence monitoring (Section 3.2.3, lines 212-213)
   - Added comprehensive explanation for handling LLM's refusal responses (Section 3.2.3, lines 220-225; Section 3.3, line 280)
3. Strengthened Related Work and Connections
   - Expanded the related work section on Knowledge Conflicts (Appendix B.1, lines 934-956)
   - Elaborated on the relationship and distinctions between Knowledge Conflicts and Adaptive RAG

These revisions have been carefully implemented to address the reviewers' concerns while maintaining the paper's clarity and scientific rigor. We believe these changes have substantially improved the manuscript's quality and hope they meet your expectations.

Thank you again for your valuable feedback. We look forward to your reply and are happy to answer any further questions.

Best,

Authors

---

### Meta-Review · Area_Chair_7yma · 2024-12-21

**Metareview:**

In this paper, the authors propose an adaptive RAG framework to address hallucination and efficiency issues. Reviewers found the idea interesting and appreciated the lightweight nature of the framework. However, they raised questions about its novelty and whether the underlying assumption—that confidence and honesty can be linearly separated in the latent space—is universally applicable, particularly for complex tasks and when using closed-source LLMs. Concerns were also expressed regarding the computational complexity of the proposed approach.

**Additional Comments On Reviewer Discussion:**

The authors provided responses to address the reviewers' concerns. While the reviewers acknowledged reading the rebuttal, they remained unconvinced.

---

### Decision · Program_Chairs · 2025-01-22

Reject